# Spatiotemporal orchestration of mitosis by cyclin-dependent kinase

Nitin Kapadia[1✉] & Paul Nurse[1,2]

Mitotic onset is a critical transition for eukaryotic cell proliferation. The commonly held view of mitotic control is that the master regulator, cyclin-dependent kinase (CDK), is first activated in the cytoplasm, at the centrosome, initiating mitosis[1–3]. Bistability in CDK activation ensures that the transition is irreversible, but how this unfolds in a spatially compartmentalized cell is unknown[4–8]. Here, using fission yeast, we show that CDK is first activated in the nucleus, and that the bistable responses differ markedly between the nucleus and the cytoplasm, with a stronger response in the nucleus driving mitotic signal propagation from there to the cytoplasm. Abolishing cyclin−CDK localization to the centrosome led to activation occurring only in the nucleus, spatially uncoupling the nucleus and cytoplasm mitotically, suggesting that centrosomal cyclin−CDK acts as a 'signal relayer'. We propose that the key mitotic regulatory system operates in the nucleus in proximity to DNA, which enables incomplete DNA replication and DNA damage to be effectively monitored to preserve genome integrity and to integrate ploidy within the CDK control network. This spatiotemporal regulatory framework establishes core principles for control of the onset of mitosis and highlights that the CDK control system operates within distinct regulatory domains in the nucleus and cytoplasm.

At mitotic onset, the cell undergoes many rapid changes across the cell, including modifications to the nuclear membrane, chromosome condensation, disassembly of cytoskeletal microtubules and formation of the mitotic spindle[9]. These changes are driven mainly by cyclin−CDK complexes[10] through phosphorylation of hundreds of different substrates[10–12]. However, a full understanding of the spatial and temporal dynamics of the CDK control network has been difficult to establish. This is in part owing to the existence of multiple different cyclin−CDK complexes and regulators in most eukaryotes, which have potential overlapping and redundant functions, making interpretation of experiments complex[12–15]. In addition, in vitro systems designed to investigate the dynamics of CDK activation that take account of cellular spatial compartmentation have been limited[5,6,14]. Finally, there is a lack of precise and simultaneous single-cell measurements of in vivo CDK activity and endogenous levels of CDK and its regulators[16,17].

The Cdk1 kinase in complex with its regulatory subunit cyclin is the main driver triggering cellular changes at mitotic entry[18]. Its activity increases gradually during the cell cycle until late G2, when it increases more rapidly to bring about the onset of mitosis through a combination of cyclin accumulation and regulation of Cdk1 tyrosine residue phosphorylation (Y15 in most eukaryotes) near its ATP binding site[4–6,19,20]. The inhibitor Wee1 kinase family phosphorylates Y15 to inactivate Cdk1, while the activator Cdc25 family dephosphorylates this site[21–24]. Cdk1 also phosphorylates Wee1 and Cdc25, respectively inactivating and activating them[25–28]. Theoretical modelling and biochemical experiments of these positive (Cdk1 and Cdc25) and double-negative (Cdk1 and Wee1) feedback loops have shown they are able to abruptly increase Cdk1 activity at mitotic entry[4–6]. Furthermore, these feedback loops can lead to bistability and hysteresis, where for a given concentration of cyclin, CDK activity can either be high if the state of the system is mitotic or low if it is in interphase[4,6,8]. This ensures that the mitotic transition is irreversible and that the interphase and mitotic states are clearly distinguished, thus preventing cells from slipping back and forth between them[4–6,29].

The eukaryotic cell contains multiple spatial compartments, the most prominent being the nucleus and cytoplasm. Such spatial complexity poses a challenge to understanding how a cell coordinates mitotic entry across the cell[7,13,16,30,31]. Previous studies have suggested that the initial mitotic CDK activation is triggered in the cytoplasm at the centrosome, the spindle pole body (SPB) in yeasts, which acts as a mitotic signalling hub that controls the timing of mitosis, from yeast to humans[1–3,16,32]. This view has been challenged by experiments using *Xenopus* egg extracts, which suggest that the nucleus might control the timing of mitosis[33,34]. Differential localization of mitotic regulators such as Wee1 (enriched in the nucleus)[35] also raise the possibility of differences in the bistable responses between the nucleus and cytoplasm[7].

*Schizosaccharomcyes pombe* (fission yeast) has been an excellent model system for understanding core principles in eukaryotic mitotic control[36]. This is partly owing to the relative simplicity of its cell cycle control architecture, with only one mitotic cyclin−CDK complex, the B-type cyclin Cdc13, complexed with the Cdk1 homologue Cdc2[37,38]. Cell cycle control can be further simplified by engineering a Cdc13−wild-type Cdc2 (Cdc2(WT)) fusion, which can drive the entire cell cycle[36]. Here we use fission yeast to reveal the underlying spatiotemporal regulatory principles of how CDK orchestrates the onset of mitosis in vivo.

[1]Cell Cycle Laboratory, The Francis Crick Institute, London, UK. [2]Laboratory of Yeast Genetics and Cell Biology, Rockefeller University, New York, NY, USA. ✉e-mail: nitin.kapadia@crick.ac.uk

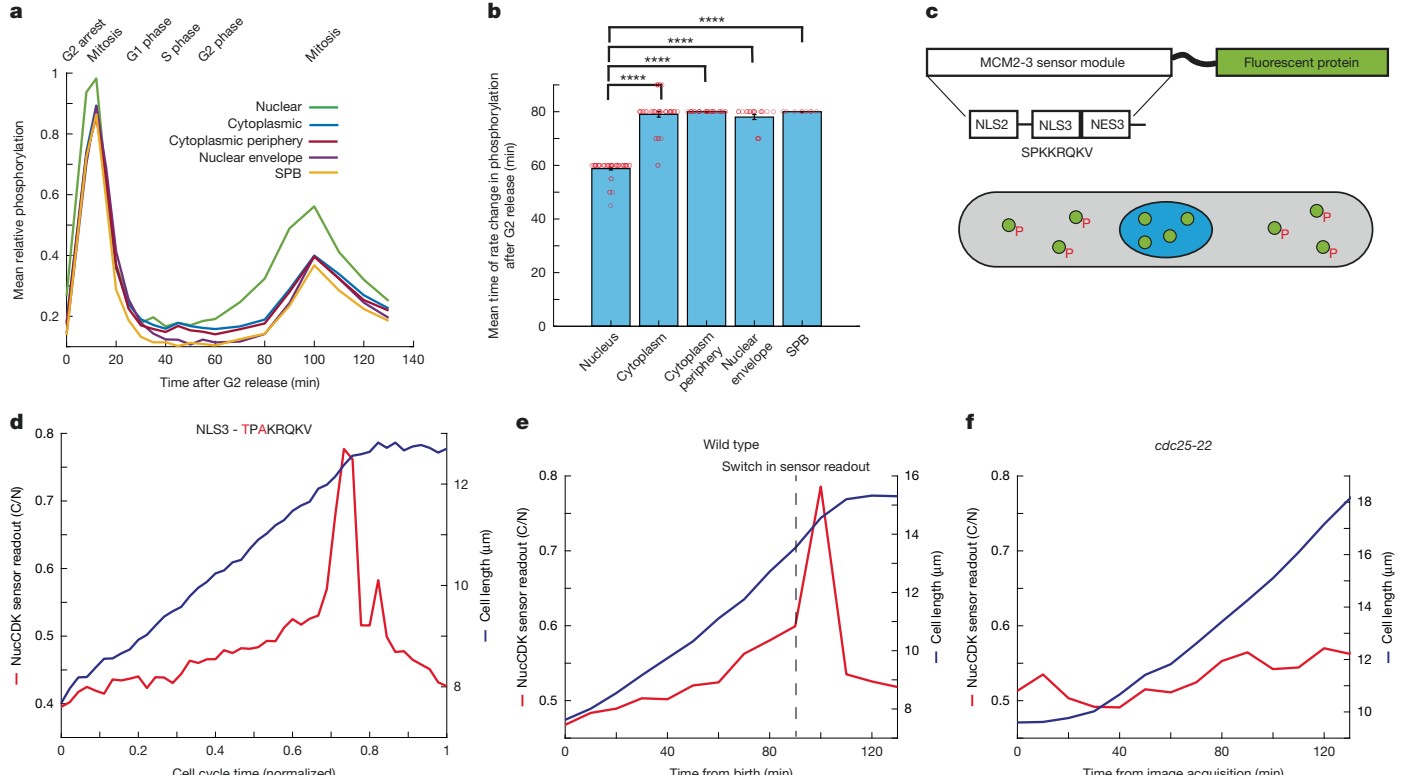

**Fig. 1 | Spatial differences in CDK substrate phosphorylation and development of CDK sensors. a**, Mean phosphorylation of CDK late sites categorized on the basis of annotated localization data. $n$ = 45 sites (nuclear), 41 sites (cytoplasmic), 27 sites (cytoplasmic periphery), 20 sites (nuclear envelope) and 7 sites (SPB). **b**, Mean times of the change points in substrate phosphorylation. An unequal two-sample, left-tailed $t$-test was performed to test whether nuclear sites had an earlier rate change compared with sites in other compartments. $P$ values: $2.92 \times 10^{-25}$ (cytoplasm), $2.90 \times 10^{-37}$ (cytoplasm periphery), $1.57 \times 10^{-18}$ (nuclear envelope) and $2.09 \times 10^{-37}$ (SPB). Error bars represent s.e.m. There was no variability with the estimated rate change points for cytoplasmic periphery and SPB sites. Red circles represent individual data points. **c**, Diagram of the minimal phosphorylation module used for construction of NucCDK. NucCDK normally resides in the nucleus (blue) but upon phosphorylation by CDK, translocates into the cytoplasm (grey). NLS, nuclear localization signal. **d**, Representative profile of NucCDK–mScarletI showing the cytoplasm to nuclear ratio of mean intensity (C/N). Cells were imaged in Edinburgh minimal medium (EMM) with 5-min time intervals. The alanine mutant of the MCM2-3 module was used as the nuclear mask (MCM2-3(Ala-mNG)). **e**,**f**, Representative single-cell trace of NucCDK readout in wild-type (**e**) and *cdc25-22* mutant (**f**) strains. Images were acquired every 10 min in YE4S medium at 36 °C.

## Spatial patterns in CDK phosphorylation

We explored whether there were differences in CDK substrate phosphorylation in different subcellular locations by reanalysing data from an earlier fission yeast phosphoproteomics experiment, to determine when CDK substrates are phosphorylated during the cell cycle[11]. We focused on sites that were phosphorylated in late G2 and mitosis using a localization annotation method[39] to categorize phosphorylation sites on the basis of their cellular compartment[11]. Sites in the nucleus increased before the cytoplasm, cytoplasm periphery, the centrosome and SPB, and the nuclear envelope (Fig. 1a,b and Extended Data Fig. 1a). There was no significant correlation between the sensitivity of the sites to CDK activity, as judged by half-maximal inhibition concentrations ($IC_{50}$ values)[11] and the timing of increases in phosphorylation at mitotic onset (Extended Data Fig. 1b). These results suggest that mitotic CDK activation occurs first in the nucleus before activation in other compartments, including the cytoplasm and SPB.

To investigate this, new single-cell sensors were required to assay CDK activity in the nucleus and cytoplasm in unperturbed cells. To assay nuclear CDK activity, we developed the CDK sensor NucCDK, which resides in the nucleus but translocates to the cytoplasm upon phosphorylation by nuclear CDK (Fig. 1c,d and Methods). NucCDK sensor readout increased gradually throughout G2, switching to a rapid increase before stoppage of cell elongation, a feature of mitotic cells[40] (Fig. 1d and Extended Data Fig. 1d–f). We tested whether this switch

occurred at mitotic onset using a temperature sensitive *cdc25-22* mutant that blocks cells in late G2 (ref. 41) at 36 °C. Unlike wild-type cells, there was no rapid switch at 36 °C, confirming that the switch was associated with mitotic onset (Fig. 1e,f). Thus, this switch provides a readout of mitotic CDK activation.

## Nuclear CDK activation drives mitosis

SynCut3 is a CDK sensor that assays cytoplasmic CDK activity[42]. It is localized in the cytoplasm in interphase but translocates into the nucleus upon CDK phosphorylation[42]. We used this sensor (hereafter referred to as CytCDK) with NucCDK, and performed dual-colour imaging to determine the timing of CDK activation simultaneously in both the nucleus and cytoplasm (Fig. 2a,b).

The readouts of NucCDK and CytCDK increased rapidly at the onset of mitosis, with the increase in NucCDK readout preceding that of CytCDK (Fig. 2c). We characterized the time delays between the lines as representing the rapid increase in sensor readouts at two points: near the beginning of the rapid increase of NucCDK and close to the peak (Methods). The time delays were about 12 min and 7 min near the beginning and near the peak, respectively, suggesting that nuclear CDK activation precedes cytoplasmic CDK activation (Extended Data Fig. 2a). To further investigate this delay, we systematically identified the initial point where the rate change occurs for each sensor readout, marking the change in CDK activity from a gradual to a rapid increase owing

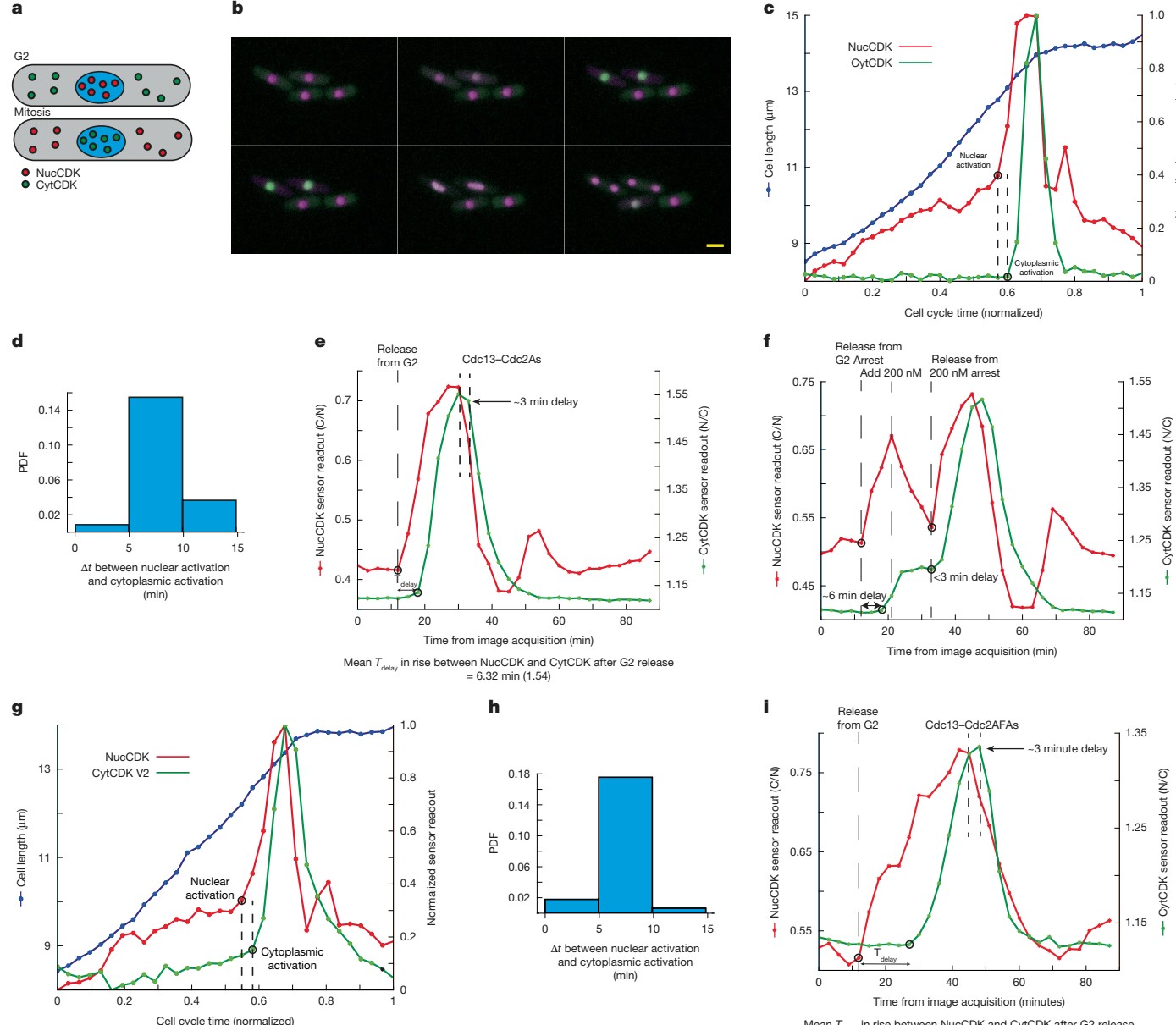

**Fig. 2 | Dual sensor system suggests that CDK activation occurs in the nucleus first. a**, Diagram of the dual sensor system. **b**, Montage with NucCDK–mScarletI (magenta) and CytCDK–mNG (green), taken every 5 min, in YE4S medium. Scale bar, 5 μm. Representative from three biological repeats. **c**, Representative trace of results from the dual sensor system. Black circles and dotted lines indicate where the rate change algorithm detected significant changes in sensor readouts. Sensor readouts were normalized to minimum and maximum values. Cell cycle time (normalized) was normalized to the division time for the individual cell. **d**, Probability density function (PDF) showing time differences from nuclear activation to cytoplasmic activation. $n = 93$ cells; 1 of 3 biological repeats. **e**, Representative trace in Cdc13–analogue-sensitive Cdc2 (Cdc2As) background from block and release from G2. DMSO was added after eight time points of image acquisition as a control. A 3-min delay (shown with dotted lines) of when sensor readouts begin to decrease owing to normal cell

cycle CDK inactivation. Mean delay time ($T_{delay}$) is shown below the graph, with s.d. in brackets. $n = 18$ cells; 1 of 2 biological repeats. **f**, Representative trace of NucCDK and CytCDK in Cdc13–Cdc2As background after release from G2, followed by inhibition with 200 nM 1-NmPP1, followed by release. **g**, Representative trace of dual sensor strain with NucCDK–mScarletI and CytCDK V2–mNG. Experiments were done in YE4S medium, with images taken every 5 min. **h**, PDF of time differences from nuclear activation to cytoplasmic activation using NucCDK–mScarletI and CytCDK V2–mNG. $n = 123$ cells; 1 of 3 biological repeats. **i**, Representative trace in the Cdc13–analogue-sensitive Cdc2AF (Cdc13–Cdc2AFs) background from block and release from G2. DMSO was added after 11 time points of image acquisition as a control. A 3-min delay (shown with dashed lines) of when sensor readouts begin to decrease owing to normal cell cycle CDK inactivation. Mean $T_{delay}$ is shown below the graph, with s.d. in brackets. $n = 30$ cells; 1 of 2 biological repeats.

to mitotic CDK activation (Fig. 2c and Methods). We observed a time delay between nuclear and cytoplasmic activation of 5–10 min, similar to the previous estimate of approximately 12 min near the beginning of the increase in NucCDK readout (Fig. 2d and Extended Data Fig. 2b). These experiments indicate that nuclear CDK activation occurs before cytoplasmic CDK activation, raising the possibility that the increase

in nuclear CDK activity might bring about the subsequent increase in cytoplasmic CDK activity.

However, the time delay at mitotic onset between the NucCDK and CytCDK readouts could be due to a delay either in CDK phosphorylation of the sensors or to differences in sensor translocation kinetics. To distinguish these possibilities, we first acutely inhibited CDK activity

after release from a G2 block during the rapid increase in CDK activity at mitotic onset, in a strain carrying an analogue-sensitive (As) version of the Cdc13–Cdc2(WT) fusion protein[36] and both NucCDK and CytCDK (Fig. 2e, Extended Data Fig. 2c,d and Methods). Upon release from the G2 block, there was a delay of around 6 min in the increase of NucCDK and CytCDK, similar to our estimate of 5–10 min (Fig. 2e). However, when cells were blocked and released during mitosis, there was no detectable delay between the increases of NucCDK and CytCDK, suggesting that both sensors translocate rapidly after CDK phosphorylation, and that the approximately 6-min delay is due to differences in the timing of CDK activation between the nucleus and cytoplasm (Fig. 2f). We also examined the response of the sensors after CDK inhibition. There was no delay between CDK inhibition and a change in NucCDK readout, but there was a delay of around 3 min in the change in CytCDK readout (Extended Data Fig. 2d). The decay half-lives for the sensors were both less than 5 min, indicating that the readouts of the sensors decay rapidly (Extended Data Fig. 2e).

We also characterized the in vivo sensitivities of NucCDK and CytCDK to CDK activity by determining the CDK inhibition dose–response curves for both sensors (Methods). This revealed that NucCDK is more sensitive to CDK activity than CytCDK (Extended Data Fig. 2f). To test whether this might affect our observation on the order of CDK activation, we developed a version of CytCDK that was more sensitive to CDK activity (CytCDK V2; Methods). CytCDK V2 had similar sensitivity to CDK activity as NucCDK (Extended Data Fig. 2g), so we repeated the dual sensor experiments. CytCDK V2 showed an earlier gradual increase in readout during G2, but the timing of the transition to a rapid increase representing CDK activation was the same as when using CytCDK (Fig. 2g,h and Extended Data Fig. 2h), confirming our intepretation of the sequential order of nuclear and cytoplasmic CDK activation.

Overall, these experiments indicate that NucCDK readout increases before that of CytCDK at mitotic onset owing to differences in the timing of CDK activation between the nucleus and cytoplasm. We performed a further experiment to determine whether nuclear CDK activation occurs sequentially before cytoplasmic CDK activation by genetically bringing about a more gradual increase in CDK activity at mitotic onset, with the assumption that at mitotic onset a critical nuclear CDK activity threshold has to be reached to trigger the subsequent increase in cytoplasmic CDK activity[17] (discussed further below). The more gradual increase in nuclear CDK activity would mean that it takes longer to reach this critical threshold, extending the delay in the increase of cytoplasmic CDK activity. We used a mutant strain carrying an analogue-sensitive version of the Cdc13–Cdc2(T14A/Y15F) fusion protein[36] (Cdc13–Cdc2AF) that abolishes the CDK Y15 feedback loops thus making the increase in CDK activity more gradual. In the strain expressing the Cdc13–Cdc2AFAs fusion, the delay in the increase of NucCDK and CytCDK was extended to 13 min after release from G2 (Fig. 2i and Extended Data Fig. 2i,j). These results further support that there is a sequential order of nuclear and cytoplasmic CDK activation, and that nuclear CDK activity has a role in controlling the timing of the increase of mitotic cytoplasmic CDK activity.

The sensors are CDK substrates that are also phosphatase substrates, which may influence the timing of the switches in sensor readout[43]. Greatwall kinase links CDK activity to PP2A–B55 phosphatase activity, so that when CDK is activated at mitotic onset, PP2A–B55 is inactivated, resulting in a rapid increase in substrate phosphorylation[43–45]. Therefore, Greatwall might influence the delay between switches in sensor readout of NucCDK and CytCDK or CytCDK V2 through phosphatase regulation. However, deletion of *ppk18*, which encodes the main Greatwall kinase in fission yeast[44], had no significant effect on the timing of the switches (Extended Data Fig. 2k).

We conclude from the sensor data and the phosphoproteomics analysis that there is a sequential temporal order in CDK activation at mitotic onset, with activation occurring first in the nucleus.

## Translocation activates cytoplasmic CDK

The translocation of cyclin B1 in human cells at the onset of mitosis is important for mitosis[16,31], so we determined whether translocation of cyclin–CDK also occurs in fission yeast, coupling cytoplasmic activation to nuclear activation. We performed timelapse microscopy of Cdc13 levels using Cdc13 tagged with superfolder GFP (sfGFP) combined with a nuclear mask to segment the nucleus (Methods). We observed a decrease in nuclear intensity of Cdc13 just before SPB separation, with a corresponding increase in the cytoplasm, indicative of nuclear export at mitotic onset (Fig. 3a and Extended Data Fig. 3a,b). This was followed by a decrease throughout the cell after spindle assembly due to Cdc13 degradation at mitotic exit (Fig. 3a and Extended Data Fig. 3a,b). For subsequent experiments, we used the mean of the top 15% of pixels as an estimate of nuclear intensity, as nuclear area comprises approximately 15% of cell area[46]. This gave similar results to using a nuclear mask (Extended Data Fig. 3c), but avoided excessive light exposure. Timelapse imaging of Cdc2–mNeonGreen (mNG) indicated a similar nuclear export happening before spindle assembly, followed by further export after Cdc13 degradation (Fig. 3b and Extended Data Fig. 3d). We performed dual-colour imaging of Cdc13–sfGFP with NucCDK and CytCDK to identify when Cdc13 export occurs in relation to nuclear and cytoplasmic activation. We found that nuclear activation occurred about 5 min before export of Cdc13, whereas cytoplasmic activation occurred concurrently with export (Fig. 3c–e). A similar 5-min difference was observed for Cdc2–mNG with respect to nuclear activation, suggesting that Cdc13 and Cdc2 translocate as a complex (Extended Data Fig. 3e). When we calculated the nuclear and cytoplasmic mean Cdc13–sfGFP intensities at nuclear and cytoplasmic activation, we found the nucleus had a higher value, indicating differences in the cyclin threshold for CDK activation between the nucleus and cytoplasm (Fig. 3f and Extended Data Fig. 3f). These differences are unlikely to be due to differences in sensor sensitivities, because the rate change point in sensor readout does not change with sensors that have different sensitivities (Fig. 2). This indicates that CDK is first activated in the nucleus, where there is a higher cyclin threshold, and this is followed by a fraction of the nuclear cyclin–CDK translocating to the cytoplasm, which overcomes the cyclin threshold in the cytoplasm to activate cytoplasmic CDK.

## Stable nuclear CDK activity oscillations

Bistability (multiple stable steady states of a system) and hysteresis (state of system dependent on its history) in Cdk1 activity has been shown in *Xenopus* egg extracts using steady-state measurements of Cdk1 activity on non-degradable cyclin[6,8]. However, to our knowledge, hysteresis in CDK activity has not been directly tested in vivo, and potential spatial differences have not been investigated in any system. Furthermore, hysteresis and positive feedback loops can be dispensable for stable oscillations in certain conditions[5]. Therefore, we determined the dynamic single-cell phase plots in the nucleus and cytoplasm in vivo, which represent the naturally occurring phase orbits in the dynamical system[5], and investigated whether hysteresis would allow translocation of Cdc13–Cdc2(WT) to the cytoplasm without collapse of the mitotic state in the nucleus. The phase plots enable us to visualize the different states of CDK activity for different concentrations of cyclin[4–6].

The phase plots using data from Fig. 3c–f indicated a significant difference in the phase orbits between the nucleus and cytoplasm (Extended Data Fig. 3g). This was confirmed using the Cdc13–Cdc2(WT) fusion protein tagged with mNG, which confines analysis to a single cyclin–CDK complex[36], together with NucCDK and CytCDK V2. The dynamics of Cdc13–Cdc2(WT)–mNG mimicked uncomplexed Cdc13 for both nuclear export and sensor readout profiles (Extended Data Fig. 4a–d). The nuclear and cytoplasmic phase orbits were plotted on the same phase space, after normalization to minimum and maximum of sensor

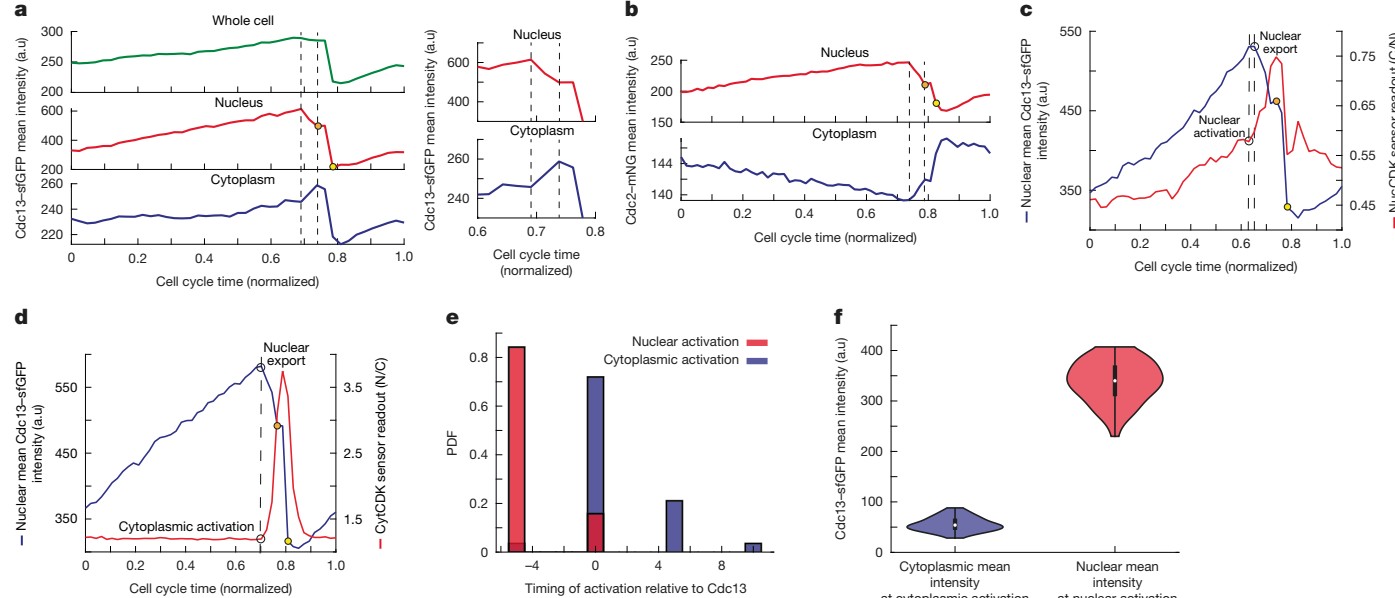

**Fig. 3 | Nuclear export of Cdc13–Cdc2(WT) is associated with cytoplasmic CDK activation. a**, Left, representative traces showing mean intensities of whole cell, nucleus and cytoplasm, using MCM2-3-Ala–mScarletI as nuclear mask. Dashed lines mark beginning and end of nuclear export. Right, zoomed-in trace. Cells were imaged in EMM, at 5-min intervals. **b**, Representative time trace of Cdc2–mNG mean intensity in nucleus and cytoplasm, calculated using the top 15% method. Dashed lines mark beginning and end of nuclear export. Imaged at 5-min intervals in EMM. **c**, Representative trace of NucCDK–mScarletI and Cdc13–sfGFP showing nuclear activation with respect to nuclear export (dashed line). Cells were imaged in EMM, with 5-min intervals. **d**, Representative trace of CytCDK–mScarletI and Cdc13–sfGFP showing cytoplasmic activation with respect to nuclear export, marked with dashed lines. Imaged at 5-min intervals in EMM. In **c**,**d**, orange circles on the nuclear mean intensity traces represent the point at which SPB separation is observed and yellow circles represent when nuclear division is observed. **e**, Histogram of nuclear (*n* = 76 cells; 1 of 3 biological repeats) and cytoplasmic (*n* = 57 cells; 1 of 3 biological repeats) activation with respect Cdc13–sfGFP nuclear export. Red represents data from NucCDK–mScarletI and Cdc13–sfGFP and blue represents data from CytCDK–mScarletI and Cdc13–sfGFP. **f**, Violin plots of nuclear mean intensity and cytoplasmic mean intensity after background subtraction, with nuclear (*n* = 76 cells) and cytoplasmic (*n* = 57 cells) CDK activation, respectively. The white dot represents the median value, the black rectangle represents the interquartile range (IQR; bounded by the 25th and 75th percentiles), and the whiskers extend to the minimum and maximum, defined as data points within 1.5× IQR. Mean background values were calculated as the mean of the median values shown in Extended Data Fig. 3f and subtracted from each value.

readouts (Fig. 4a). Notably, the nuclear phase orbit cycled around a much larger region of phase space than the cytoplasm. A high amount of Cdc13–Cdc2(WT) accumulated in the nucleus before CDK activation to the high-activity mitotic state, with little increase in activity before that point. There was an approximately vertical line on the far right of the orbit at the point of CDK activation, consistent with an underlying bistable trigger. In the nucleus, once the system had reached the mitotic high CDK activity state, it was able to tolerate decreases in Cdc13–Cdc2(WT) without collapsing back to the low-activity state. By contrast, only a small amount of Cdc13–Cdc2(WT) was needed in the cytoplasm for CDK activation, and a small decrease in Cdc13–Cdc2(WT) resulted in collapse out of the cytoplasmic mitotic state (Fig. 4a). Overall, the phase orbit in the cytoplasm was much narrower compared to the one in the nucleus. This suggests that CDK activity in the nucleus is much more stable with respect to cyclin–CDK fluctuations compared with CDK activity in the cytoplasm. The individual nuclear phase plots extended and shrunk significantly horizontally compared with the cytoplasm, also suggesting less sensitivity to cyclin–CDK concentrations (Fig. 4b).

Single-cell phase plots of the nucleus after normalization to minimum and maximum of Cdc13–Cdc2(WT)–mNG mean intensities indicated a considerable gap between the low-activity and high-activity states, and after nuclear export the system was not prone to collapse back to the low-activity state (Fig. 4c). Furthermore, the orbits were very similar in different single cells, suggesting the CDK regulatory system oscillates with robustness to biological noise. By contrast, the normalized cytoplasm phase plots showed that the gaps between the high-activity and low-activity states for the same Cdc13–Cdc2(WT)–mNG intensity were less pronounced, and the cytoplasm was more sensitive to Cdc13–Cdc2(WT) accumulation (Fig. 4d).

We repeated the experiments using Cdc13–Cdc2AF–mNG[36]. This tested whether CDK Y15 feedback loops are responsible for the stability in oscillations and for the distinction of the low and high CDK activity states in the nucleus and cytoplasm[5]. Nuclear export appeared less switch-like, suggesting that it is influenced by the altered CDK activity regulation (Extended Data Fig. 4e,f). The sensors displayed a linear increase in sensor readout (Extended Data Fig. 4g,h), and the phase diagrams of both the nucleus and cytoplasm exhibited almost a complete collapse of their orbits, with approximately linear profiles (Fig. 4e,f and Extended Data Fig. 4i,j). Thus, the CDK Y15 feedback loops promote oscillations between different states in both the nucleus and cytoplasm. In the nucleus, when export occurs, the system is subsequently prone to collapse, indicating that in the Cdc13–Cdc2AF strain, translocation comes with the risk of the nucleus collapsing back to the interphase state (Fig. 4e). We conclude that the CDK Y15 feedback loops provide stable oscillations in the nucleus, facilitating translocation of cyclin–CDK from the nucleus to the cytoplasm, and produce a weaker response in the cytoplasm. This is likely to reflect the underlying steady-state hysteretic responses that differ between the nucleus and cytoplasm[5]. Furthermore, our data suggest that at mitotic onset, the CDK Y15 feedback loops are major drivers in hysteresis and the increase in substrate phosphorylation, rather than phosphatase inactivation[43].

## Centrosomal cyclin–CDK relays activity

Our results do not support the view that mitotic CDK activation is triggered first in the cytoplasm at the centrosome or SPB thereby dictating mitotic onset timing[1,3,32]. Therefore, we investigated the possible role of cyclin–CDK at the SPB at mitotic onset. In fission yeast, CDK activation

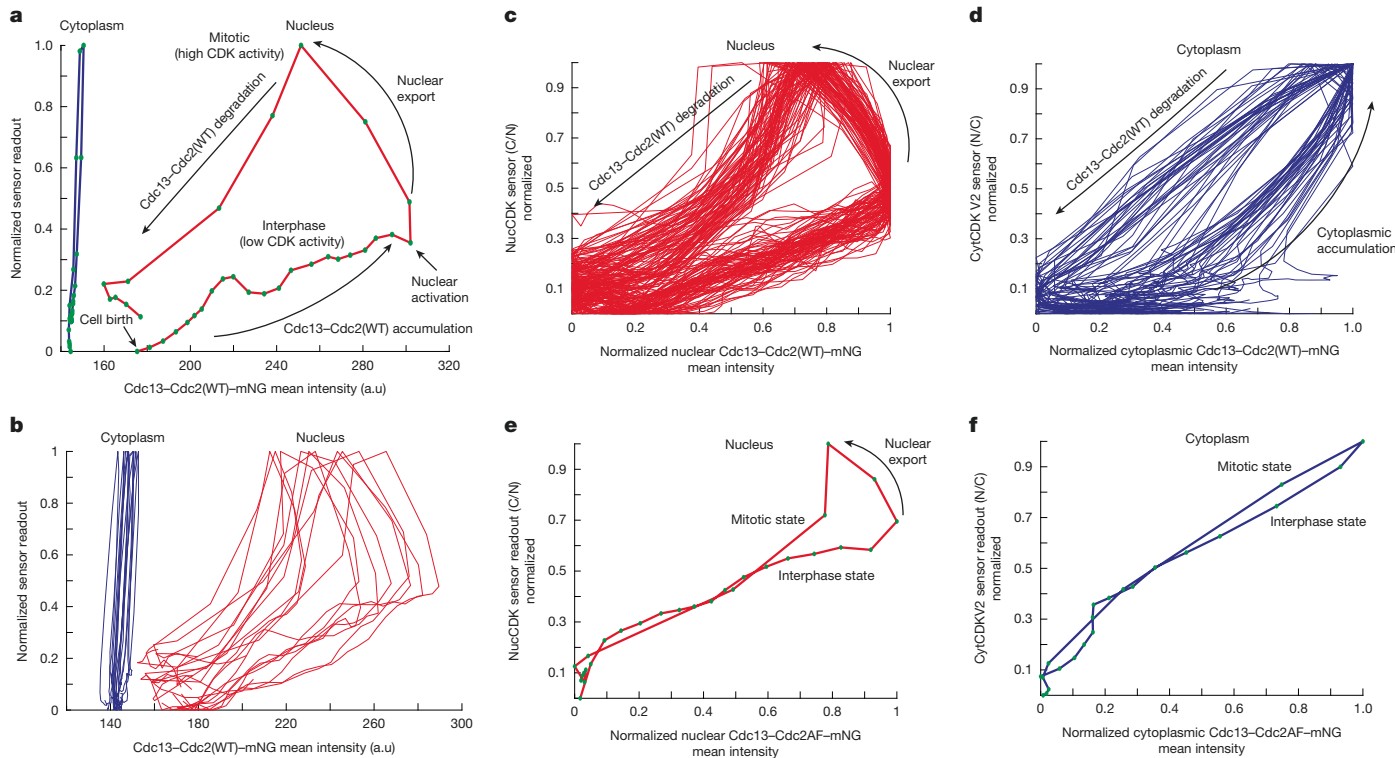

**Fig. 4 | CDK oscillatory dynamics in the nucleus and cytoplasm.**
**a**, Representative phase plots of the nuclear phase orbit (Cdc13–Cdc2(WT)–mNG nuclear mean intensity and NucCDK–mScarletI) and cytoplasmic phase orbit (Cdc13–Cdc2(WT)–mNG cytoplasmic mean intensity and CytCDK V2–mScarletI). Sensor readouts were normalized to minimum and maximum values for comparison. **b**, Ten representative traces for nucleus and cytoplasm. **c**, Phase plots for the nucleus in Cdc13–Cdc2(WT)–mNG background. Normalized to minimum and maximum values. *n* = 89 cells;

1 of 3 biological repeats. **d**, Phase plots for the cytoplasm in Cdc13–Cdc2(WT)–mNG background. Normalized to minimum and maximum values. *n* = 37 cells; 1 of 3 biological repeats. **e**, Representative phase plot for the nucleus in the Cdc13–Cdc2AF–mNG background. Normalized to minimum and maximum values. **f**, Representative phase plot for the cytoplasm in the Cdc13–Cdc2AF–mNG background. Normalized to minimum and maximum values. Green dots represent individual time points, with distances separated by a time interval of 5 min.

at the SPB has been shown to act through a polo kinase (Plo1)-mediated feedback loop, dependent on Plo1 association with the SPB scaffold protein Cut12 (refs. 1,47–50) (Fig. 5a). The mutant *cut12.s11* promotes earlier recruitment of Plo1 to the SPB and global Plo1 activation[48,49], so we determined whether *cut12.s11* advanced the timing of nuclear CDK activation (Fig. 5b). Cell length is a proxy for cell cycle position in fission yeast[51]. Length at nuclear CDK activation assayed using NucCDK (Fig. 5c, left) and cell length at cell division (Fig. 5c, right), were found to be the same in both *cut12.s11* and wild-type cells. This indicates that promoting premature CDK activity at the SPB does not affect either nuclear CDK activation or the timing of cell division. This is in line with previous reports that promoting SPB feedback under a normal cell cycle context does not influence mitotic timing[47].

We next investigated the role of cyclin–CDK localization at the SPB for mitotic control using the hydrophobic patch mutant of Cdc13 (Cdc13(HPM)), which does not localize to the SPB during G2[39] (Fig. 5d). This property is conserved with mammalian cyclin B1(HPM), which also does not localize to the centrosome[39]. Cells containing only Cdc13(HPM) do not display signs of mitotic entry, a phenotype partly rescued by tethering Cdc13(HPM) back to the SPB[52]. We performed timelapse microscopy of Cdc13(HPM)–sfGFP and wild-type Cdc13 (Cdc13(WT))–sfGFP using the CDK sensors (Methods) in a strain with other cyclins deleted. Cdc13(WT)–sfGFP displayed normal sensor readouts and Cdc13 dynamics (Extended Data Fig. 5a–c). NucCDK in the Cdc13(HPM)–sfGFP mutant showed that activation in the nucleus still occurred, even though Cdc13(HPM)–sfGFP did not localize to the SPB (Fig. 5e and Extended Data Fig. 5d,e). However, this nuclear CDK activation was not followed by a decrease in nuclear levels, indicating that export from the nucleus did not take place (Fig. 5f).

We repeated the timelapse experiments with Cdc13(HPM)–sfGFP and CytCDK and found that cytoplasmic CDK activation was also absent (Fig. 5g and Extended Data Fig. 5f). Therefore, although nuclear activation still occurs, cytoplasmic CDK activation is impaired, a probable consequence of Cdc13(HPM)–Cdc2(WT) not being exported from the nucleus.

We propose that Cdc13 localization to the SPB is not required for CDK activation in the nucleus, but is required for the translocation of Cdc13–Cdc2(WT) to the cytoplasm and subsequent activation of cytoplasmic CDK. These results are consistent with previous phosphoproteomics data[39], which showed phosphorylation of nuclear substrates with Cdc13(HPM) was similar to that with Cdc13(WT), whereas phosphorylation of SPB and cytoplasmic substrates was impaired. They also support the sequential ordering of nuclear activation and cytoplasmic activation, with Cdc13 SPB localization having an important role in bringing about this ordering of events.

## Discussion

Here we have revealed the underlying spatiotemporal regulatory framework in fission yeast that determines how CDK orchestrates the onset of mitosis in vivo (Fig. 5h), based on precise measurements of cyclin–CDK levels, the sensing of in vivo CDK activity and the generation of in vivo phase plots, to investigate the effects of cellular compartmentation in single unperturbed cells.

Our work has shown that CDK is first activated in the nucleus, thereby controlling the timing of mitotic onset. Promoting activity at the SPB had no effect on the timing of nuclear CDK activation or timing of cell division. However, in the absence of cyclin B–CDK localization to the

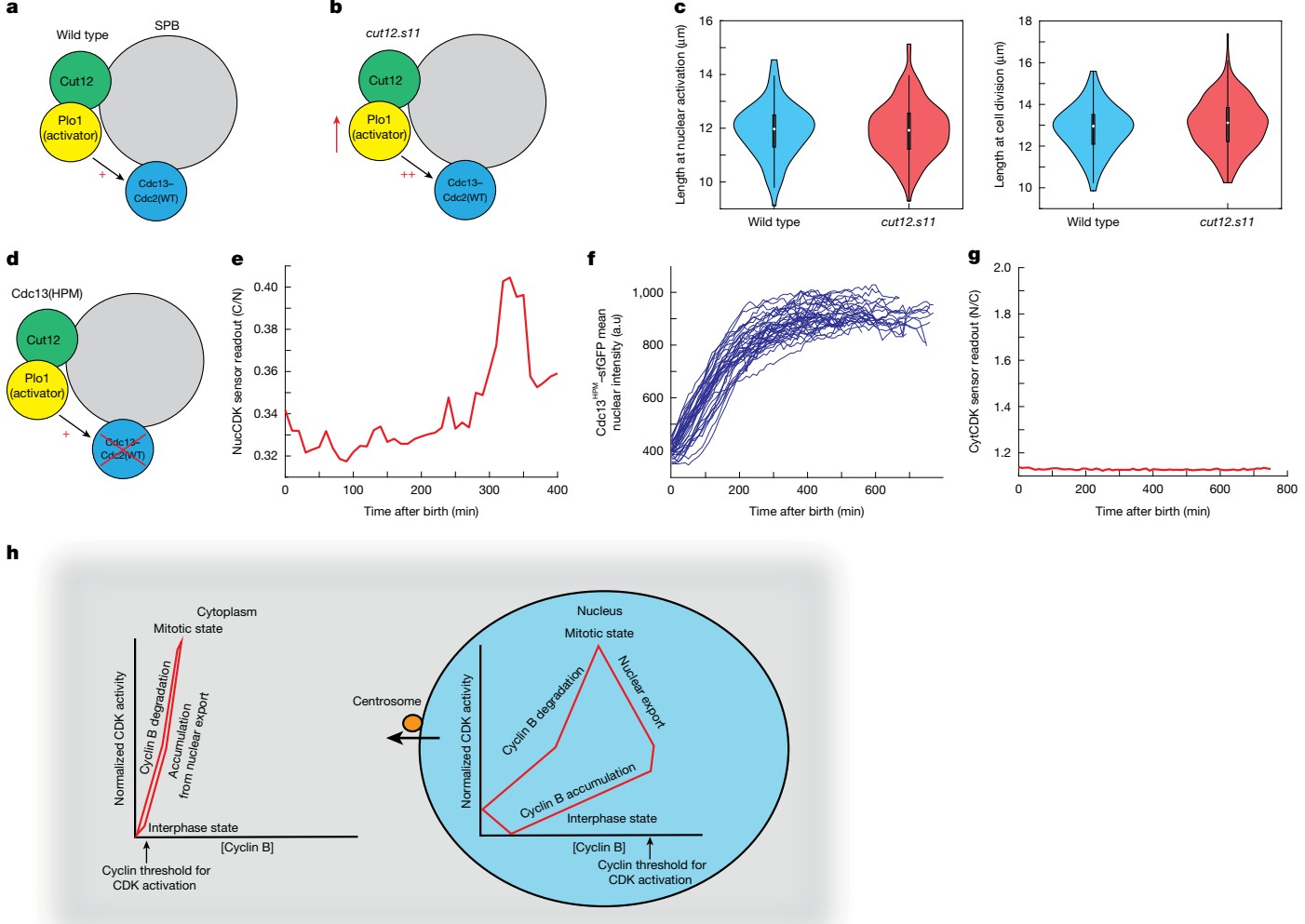

**Fig. 5 | Cdc13–Cdc2(WT) localization to the SPB does not affect nuclear CDK activation. a,b**, Diagram of the SPB in a wild-type background (**a**) and in the *cut12.s11* mutant. **c**, Left, wild-type (*n* = 96 cells; 1 of 3 biological repeats) and *cut12.s11* (*n* = 195 cells; 1 of 3 biological repeats) cells, showing length at nuclear activation using NucCDK–mScarletI. Right, length of wild-type (*n* = 96 cells) and *cut12.s11* (*n* = 195 cells) cells at cell division. The white dot represents the median value, the black rectangle represents the IQR, and the whiskers extend to the minimum and maximum, defined as data points within 1.5× IQR. **d**, Diagram of the SPB in the *Cdc13^HPM* background. **e**, Representative trace of NucCDK–mScarletI in cells expressing Cdc13(HPM)–sfGFP. **f**, Traces of mean nuclear Cdc13(HPM)–sfGFP intensity. *n* = 36 cells; 1 of 3 biological repeats. **g**, Representative trace of CytCDK–mScarletI in cells expressing Cdc13(HPM)–sfGFP. The *y* axis of CytCDK sensor readout was scaled on the basis of minimum

and maximum values for CytCDK in Cdc13 wild-type cells. **h**, Model of CDK activation. During interphase, cyclin B–CDK accumulates in the nucleus, and CDK activity increases gradually, with a high threshold for activation. In late G2, CDK activation occurs and is amplified by CDK-Y15-dependent feedback loops. Subsequently, cyclin B–CDK nuclear export (dependent on centrosomal cyclin B–CDK) and cyclin B degradation occur, and high CDK activity is maintained by CDK-Y15-dependent feedback loops. Thus, the nuclear CDK system undergoes stable oscillations with respect to cyclin B–CDK concentration. By contrast, in the cytoplasm, a lower threshold for activation enables cytoplasmic CDK to activate upon import of cyclin B–CDK from the nucleus. Degradation of cytoplasmic cyclin B results in an immediate drop in CDK activity. Therefore, the cytoplasmic CDK system oscillates in an unstable manner with respect to cyclin–CDK concentration.

SPB, CDK activation occurs only in the nucleus. This spatially uncouples the onset of mitosis, with the nucleus entering the mitotic state, whereas the cytoplasm remains in a G2-like state. We propose that this is owing to the absence of signal propagation brought about by nuclear export of cyclin B–CDK to the cytoplasm. This stresses the importance of signal propagation for spatially coherent mitotic onset, and supports the view that the SPB acts as a signalling hub, but as a signal-relaying hub rather than one that controls the timing of mitotic onset.

We also generated novel in vivo single-cell phase plots of CDK activity versus concentrations of cyclin B and of cyclin B–CDK complexes simultaneously in both nucleus and cytoplasm. These showed that the CDK oscillator operates in very different regulatory domains in the nucleus and cytoplasm. In the nucleus, there is a strong bistable or hysteretic response, which can buffer against fluctuations in cyclin B–CDK concentrations in both interphase and mitosis, resulting

in stable oscillations of CDK activity. By contrast, the response in the cytoplasm is much weaker. The strong bistable response and higher cyclin B threshold for CDK activation in the nucleus reduces susceptibility to 'noise' and to the risk of slipping in and out of mitosis, helping to maintain genomic stability[5,29]. This regulation could be owing to the higher concentrations of CDK regulators in the nucleus such as Wee1 and Cdc25 (refs. 7,35,46). By contrast, the less well stabilized CDK circuit and lower cyclin B threshold in the cytoplasm allows cytoplasmic CDK to respond rapidly to the translocation of cyclin B–CDK from the nucleus. Therefore, the cytoplasm may exist in a more fluid state, where the distinction between interphase and mitosis is reduced. The translocation of cyclin B–CDK links the different oscillatory orbits between the nucleus and cytoplasm, leading to coherent mitotic entry, with the nucleus enforcing control over the cytoplasm[7]. Theoretical modelling of the metazoan mitotic network[7] has proposed differences

between the nucleus and cytoplasm in bistable responses and cyclin B thresholds similar to those that have been shown here experimentally. Bistability is a common characteristic of biological signalling networks and cell transitions[6,53,54], and we have shown that the responses can be very different between cellular compartments.

Modelling of the mammalian mitotic network has suggested that nuclear envelope breakdown (NEBD) creates a potential 'stress' at mitotic entry[55], owing to dispersion of cyclin B1–Cdk1 and Greatwall kinase, resulting in mitotic collapse—a collapse that is prevented by Cdk1-Y15 feedback loops, which retain high CDK activity[55]. The nuclear export of cyclin B–CDK observed here in fission yeast can potentially lead to a similar stress in the nucleus. This is also prevented by CDK Y15 feedback loops which help to retain high CDK activity, ensuring that nuclear mitotic events such as chromosome condensation and mitotic spindle formation continue. By contrast, the cytoplasm is ill-suited for propagating CDK activity because of its weaker bistable response. We propose that the strong bistable response in the nucleus allows propagation of CDK activity from the nucleus to the cytoplasm without collapse of nuclear CDK activity. Although the fission yeast cyclin B–CDK(T14A/Y15F) (CDKAF) is viable, it exhibits sluggish mitotic progression and aberrant timing of mitotic entry, and generates a higher frequency of inviable cells[36,42].

The fission yeast CDK regulatory framework can be expected to be relevant to other eukaryotes. The *Xenopus* extract studies proposing that the nucleus is the 'pacemaker' for the cell cycle[33,34] suggest that our observations are not simply due to differences between a closed versus open mitosis. In human cells, the mechanism behind cyclin B1 nuclear translocation before NEBD is thought to be due to either Cdk1 phosphorylation of transport machinery or through a spatial positive feedback mechanism based on cyclin B1 autophosphorylation whereby the nucleus promotes the translocation of cyclin B1[16,31]. This translocation is thought to be important for triggering NEBD[17]. In fission yeast, the translocation of Cdc13–Cdc2(WT) from the nucleus to the cytoplasm could be functionally anagolous to translocation of cyclin B1–Cdk1 triggering NEBD. Therefore, although the precise mechanisms may lead to specific differences, the general concepts of the nucleus acting as the mitotic pacemaker and translocation being an important aspect for the onset of mitosis may be conserved[16,33,34].

Why does the initial mitotic CDK activation and strong bistable or hysterisis response occur in the nucleus? A major vulnerability of the mitotic cell cycle is the risk of premature CDK activation and mitotic onset when DNA is incompletely replicated or has unrepaired damage[56]. Having the CDK regulatory system within the nucleus, positions the bistable circuit close to DNA, so incomplete DNA replication and DNA damage can be readily integrated with control of CDK activity[57,58]. Such a tight association between DNA and CDK activation would help to ensure genome integrity. It also enables genome content (ploidy) to be more easily monitored, which is necessary given that cell size at mitosis increases with ploidy[42]. Thus, DNA may serve as a platform for control of CDK activity and the initiation of mitotic entry[42,57].

The principles we that have developed here will be a useful guide for future experiments in the more complex metazoan cells for investigating how CDK brings about cell cycle progression. Furthermore, the signalling network in the CDK oscillator system has been shown to have similar features to other biological oscillators, such as bistability and positive feedback loops[59]. Therefore, our work here should also be useful for understanding how other biological oscillators operate across space and time in vivo.

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

## Methods

### Development of NucCDK and CytCDK sensors

Sensors were expressed under the low expression constitutive calmodulin (cam1) promoter[60]. The sensors were tagged with either mNeonGreen or mScarletI[61,62]. NucCDK was derived from a minimal budding yeast CDK-dependent transport module, which was shown to be exported out of the nucleus after phosphorylation by Cdc28 (budding yeast Cdk1)[63] (Fig. 1c). This sensor functions similarly to a CDK2 translocation-based sensor, based on a fragment of human DNA helicase B fragment (DHB), which translocates to the cytoplasm upon CDK2 phosphorylation[64]. The MCM2-3 sensor module displayed a readout patten that increased at S phase, similar to that of early CDK substrates, consistent with observations in budding yeast[11,63] (Extended Data Fig. 1c). Although this version of the sensor is useful for the G1/S transition given its high sensitivity to CDK activity[65], we aimed to reduce its sensitivity so that it can be used to read out activity at G2/M. The module contains a full CDK consensus site beginning with a serine, next to the Mcm3 NLS. Mutation of this residue to alanine (MCM2-3 Ala) reduced the readout of the sensor and rendered it prominently nuclear, albeit with a modest increase after stoppage of cell elongation (Extended Data Fig. 1d). Therefore, we hypothesized that mutating this serine to a threonine could alter the sensitivity of the sensor as it has been shown that CDK has a greater specificity towards serine residues than threonine residues, and, furthermore, that threonines are preferentially dephosphorylated by PP2A[66,67]. As shown in Extended Data Fig. 1e, this mutation markedly altered the profile of the sensor, and gave a readout of CDK activity throughout the cell cycle. We refer to this sensor as NucCDK. The profile was very similar to a recent CDK Forster resonance energy transfer sensor tagged with a NLS, developed in fission yeast, indicating that NucCDK is biased towards sensing nuclear CDK activity[68]. We did not notice a significant difference in the pattern when using a nuclear mask for determining the cytoplasmic to nuclear ratio in mean intensities, as opposed to using the mean of top 15% of pixels and bottom 85% of pixels as estimates for nuclear and cytoplasmic mean intensities, respectively, as has been previously used in fission yeast[46] (Extended Data Fig. 1e,f). Unless stated that a nuclear mask was used, the top 15% and bottom 85% method was used for sensor readout measurements.

Syncut3 (CytCDK) translocation into the nucleus is dependent on phosphorylation of T19 by CDK[42]. CytCDK V2 was developed in a similar manner to NucCDK except T19 was mutated to a serine, for increased sensitivity to CDK activity.

### Phosphoproteomics timecourse analysis

The timecourse analysed for Fig. 1a was from the Cdc13–Cdc2(WT) fusion experiment in which Cig1, Cig2 and Puc1 had been deleted (ΔCCP), and cells grown in EMM4S + SILAC (stable isotope labelling with amino acids in cell culture) medium at 32 °C (ref. 11). Localization annotation for CDK sites was implemented as previously described[39]. In brief, the order of determining the localization of a protein by priority based on existing literature was as follows: (1) direct visualization of fluorescently tagged proteins labelled at the endogenous locus; (2) fluorescently tagged proteins from exogenous nmt41 or nmt81 promoters; (3) indirect methods suchs as chromatin immunoprecipitation assays, fractionated western blotting and immunoprecipitation experiments. Only sites annotated as being late sites as well as belonging to solely one spatial compartment were used for analysis.

### Growth and strain construction

Strains were grown, and cell cultures and media were prepared as described[69]. Strains were constructed either by transformation or genetic crossing, as described[69]. All experiments were done in exponentially growing cells at $2.5–4.0 \times 10^6$ cells per ml. Insertions were checked with colony PCR and sequencing in the case of deletions. All strains and plasmids used are listed in Supplementary Table 1. EMM powder (MP Biomedicals) with ammonium chloride and dextrose was used, supplemented with adenine at final concentrations of $0.15\ \text{g l}^{-1}$ where required. Where yeast extract with adenine, leucine, histidine and uridine (YE4S) was used, supplements were added at $0.15\ \text{g l}^{-1}$ final concentrations. The Cdc13 strain internally tagged with sfGFP is described in ref. 70.

### Microscopy

Except for some data in Figs. 2 and 4, microscopy was done on 1% agarose pads, using GeneFrames, to prevent drying of pad during acquisition[71] (125 µl, Thermofisher AB-0578). N-propyl gallate (Sigma, 02370) was added to the agarose pads at 0.1 mM final concentration to prevent photobleaching[72].

A 1-ml culture of cells was spun down once at 2,000 rpm (Eppendorf Centrifuge 5424) for 30 s, to minimize stress. Approximately 50 µl of supernatant was left and mixed. Two microlitres of cell culture was spread across an agarose pad and left to dry for a few minutes. A coverslip (Epredia, 22 × 32 mm, no. 1.5) was placed onto the pad, but only pressure along the gene frame was applied to seal up the frame. Nail polish was added along the frame to ensure that the coverslip and frame were sealed.

Imaging was done on a Nikon Ti2 inverted microscope equipped with a perfect focus system (PFS), an Okolab environmental chamber, and a Prime BSI sCMOS camera (Photometrics). A Nikon 100x Plan Apo oil immersion lens with a NA of 1.45 was used for imaging. Image acquisition was controlled using µManager V2.0 software[73]. 2 × 2 pixel binning was done to improve fluorescence signal detection with a resulting pixel size of 130 nm × 130 nm. Unless stated otherwise, imaging was done at 25°. Except for data shown in Extended Data Fig. 2a, where 5 z-slices were collected, 7 slices were collected with 0.5-µm step sizes. Fluorescence excitation was done using a SpectraX LED light engine (Lumencor). Imaging for mNG-tagged proteins was done with the YFP channel (both in excitation and emission) to minimize the influence of cell autofluorescence[74]. We did not notice any issues with growth or cell length at division under the imaging conditions used.

### Cdc13^HPM experiments

Cells contained a copy of $Cdc13^{WT}$ or $Cdc13^{HPM}$ expressed under the native Cdc13 promoter and a second copy of $Cdc13^{WT}$, expressed under the thiamine-repressible nmt41 promoter[39]. Cells were initially grown in EMM without thiamine but for experiments, cells were washed once in EMM + thiamine and placed on a 1% EMM + thiamine agarose pad. Thiamine was added to EMM at a final concentration of 30 µM. Only cells that were born around the beginning of acquisition and blocked at the end of acquisition were considered for analysis. The total acquisition was 80 time points with time intervals of 10 min, for around 13 h in total in acquisition.

### Data and image analysis

Unless otherwise stated, all analysis was done using custom scripts written in Matlab R2022a.

For processing and analysis of fluorescence z-stacks, maximum intensity projections were used using Fiji[75]. Cells were segmented using a custom algorithm as described[52]. Nuclear masks were obtained with Fiji, using MCM2-3 Ala to identify the nucleus. Cells were tracked using LineageMapper[76]. With the exception of cases where mutants that block cells in G2 were used, only cells that were born and that divided within the timelapse acquisition period were used for analysis.

For histograms, the left edges of bins are inclusive, while the right edges are not inclusive except for the last bin. Bin widths were set to

the time interval used for image acquisition, with the exception of Fig. 3e. Violin plots display distributions as kernel density estimates. The median value is plotted as a white circle. The black rectangle represents the IQR, and the black solid line represents 1.5× IQR.

### Rate change analysis

For Extended Data Fig. 2a, the time delay between the lines in the rapid rise of the NucCDK and CytCDK readouts were measured at the point of nuclear CDK activation, indicated by a change in the rate of increase of the sensor readout, and around 75% of the activation peak.

For all other figures, points representing CDK activation were identified using an automated rate change analysis algorithm. Sensor traces were lightly smoothed using the Matlab function, smoothdata and the algorithm sgolay to assist in identifying change points. However, plots in figures represent the raw data. To confine analysis to the period in G2 where the rate change in sensor readout occurs, the trace was cropped to 10–15 time points before the maximum peak in sensor readout, up to the maximum peak. The abrupt changes in sensor readout were identified using a change point algorithm[77,78] in Matlab called findchangepts, where abrupt changes in the slope were identified using the specifier 'linear'. A similar process was followed to identify points of stoppage of cell elongation.

For the phosphoproteomics analysis to determine statistical significance of nuclear substrate phosphorylation compared to the other compartments, the rate change analysis was similarly performed on individual sites.

### Sensor translocation control data experiments and sensitivity measurements

We used an analogue-sensitive sensitive version of Cdc2, Cdc2as, that renders it susceptible to inhibition by the ATP analogue 1-NmPP1[79], to determine how quickly the sensors respond to changes in CDK activity. Experiments were performed on soybean lectin-coated MatTec dishes in YE4S medium at 32 °C, with 3-min intervals[42]. Cells were blocked in G2 for 3 h using 1 μM 1-NmPP1 and released following 3× washes using YE4S medium. Subsequently, either 10 μM 1-NmPP1 was added to the dish to inhibit CDK activity or DMSO (control) was added. The mean decay curves were fitted using either a two-exponential model (NucCDK) or a one-exponential model (CytCDK) to determine the half-lives of decay.

Dose–response curves of NucCDK and CytCDK/CytCDK V2 to CDK inhibition were done similarly as described[11,42]. We used the Cdc13–Cdc2AFas strain to remove potential non-linear responses in phosphorylation of the sensors due to CDKY 15 feedback loop activation. Cells were blocked at 32 °C in YE4S for 1.5 generations. They were washed (3×) and released into release medium containing varying concentrations of 1-NmPP1. Image acquisition was done after 12 min from the first wash on microscope slides.

### Phase plots

The mNG tag was used at the C terminus of the Cdc13–Cdc2(WT) fusion, as it has better photophysical properties compared with sfGFP and allowed use of the YFP channel, which is less affected by cellular autofluorescence[61,74]. Experiments for Fig. 4 and Extended Data Fig. 4 were done using the CellASIC ONIX V1 microfluidics platform with Y04C microfluidics plates (Merck) at 32 °C in YE4S, as they are the optimal conditions used for Cdc13–Cdc2AF[80]. All 4 experimental conditions—Cdc13–Cdc2(WT)–mNG + (NucCDK–mScarletI or CytCDK V2–mScarletI) and Cdc13–Cdc2AF–mNG + (NucCDK–mScarletI or CytCDK V2–mScarletI)—were done on the same plate using a flow rate of 2.5 pound per square inch (psi). Fifty microlitres of culture at an optical density of 0.2–0.3 was added to plate. Images were acquired every 5 min.

Phase plots shown in figures were plotted after lightly smoothing the raw data using the matlab function smoothdata with the

algorithm sgolay to reduce noise and improve the quality of the phase plots.

### Reporting summary

Further information on research design is available in the Nature Portfolio Reporting Summary linked to this article.

## Data availability

All strains and plasmids available upon request. Owing to the size of the timelapse imaging datasets, we have stored these at the Francis Crick Institute and have made them available upon request by contacting the authors.

## Code availability

Matlab scripts used for image and data analysis can be found at: https://github.com/nkapadia27/Spatiotemporal-Orchestration-of-Mitosis and https://doi.org/10.5281/zenodo.11072087 (ref. 81).

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

**Acknowledgements** The authors thank T. Zeisner, J. Curran, and J. Greenwood for comments on the manuscript. This work was supported by the Francis Crick Institute, which receives core funding from Cancer Research UK (CC2003), the UK Medical Research Council (CC2003), and the Wellcome Trust (CC2003). For the purpose of Open Access, the author has applied a CC-BY copyright licence to any author accepted manuscript version arising from this submission. In addition, this work was supported by a Wellcome Trust grant to P.N. (grant number 214183) and the Breast Cancer Research Foundation (BCRF-23-117). N.K. was supported by an EMBO postdoctoral fellowship (EMBO ALTF 705-202) and a HFSP postdoctoral fellowship (HFSP LT000587/2021-L).

**Author contributions** N.K. initiated the study. N.K designed and performed all the experiments. N.K. generated the strains and plasmids. N.K performed the image and data analysis including writing custom scripts. N.K. and P.N. wrote the manuscript.

**Funding** Open Access funding provided by The Francis Crick Institute.

**Competing interests** The authors declare no competing interests.

**Additional information**
**Correspondence and requests for materials** should be addressed to Nitin Kapadia.

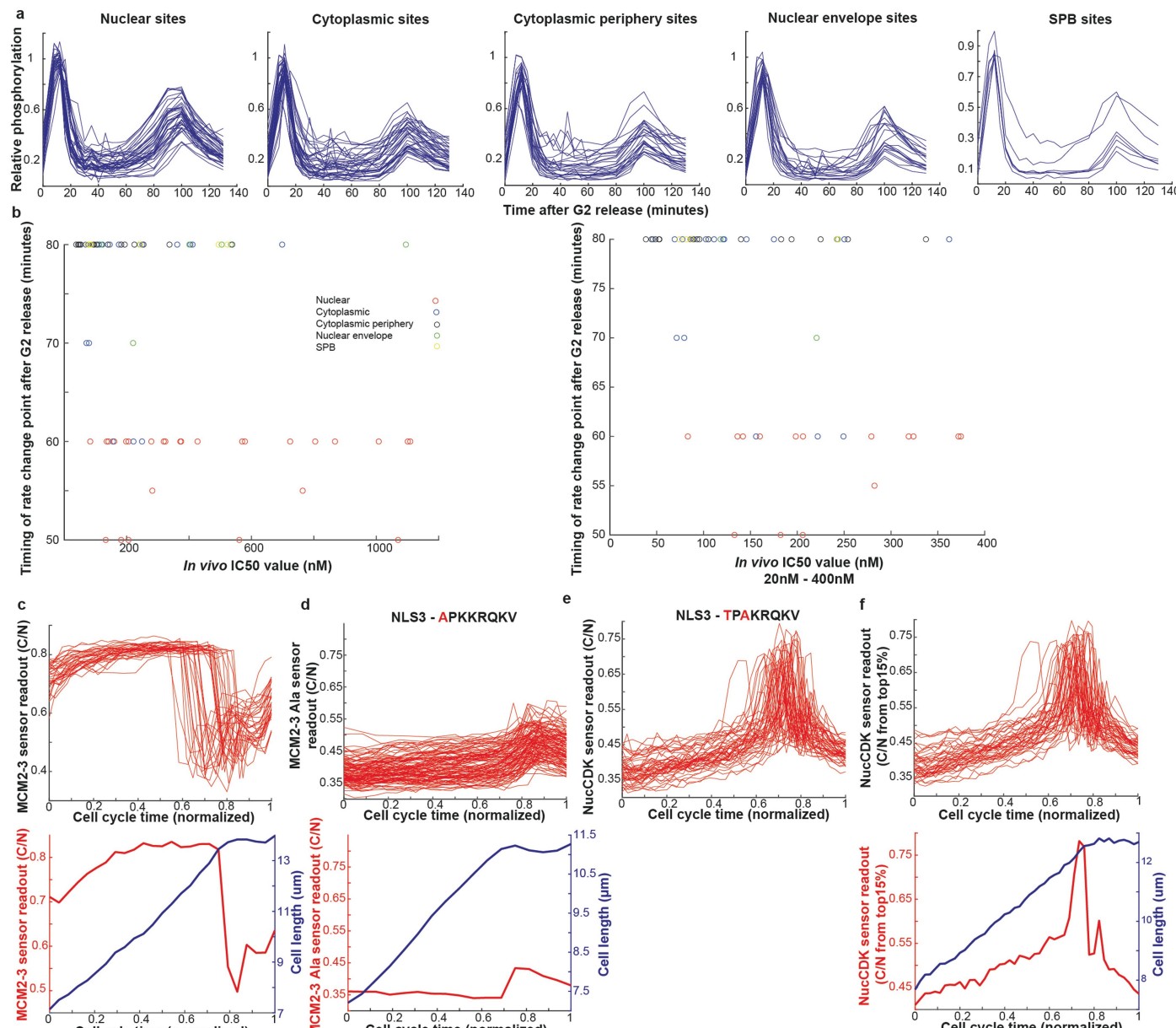

**Extended Data Fig. 1 | Timing of rate change point is associated with subcellular compartment, and mutations affect CDK sensor sensitivity.**
a) Profiles of individual late sites in the nucleus (n = 45 sites), cytoplasm (*n* = 41 sites), cytoplasmic periphery (*n* = 27 sites), nuclear envelope (*n* = 20 sites), and SPB (*n* = 7 sites). b) left) Plot of IC50 value vs rate change point after G2 release (minutes). Spatial compartments are colour coded as legend shows. Right) Plot as left but scaled to show IC50 values in the range of 20 nM to 400 nM. c) Top: Traces of the original MCM2-3 sensor tagged with mScarletI, using the cytoplasmic to nuclear ratio as a measure of sensor readout, taken every 10 min, in EMM media. The mean top 15% of pixels were used as a measure for

mean nuclear intensity, while the mean of bottom 85% of pixels was used as a measure for mean cytoplasmic intensity. *n* = 35 cells; 1 of 3 biological repeats. Bottom: Representative time trace. d) Top: MCM2-3-Ala-mScarletI traces, taken every 5 min, in EMM media. Nuclear masks were obtained using MCM2-3-Ala-mScarletI. Y-axis scaled to MCM2-3 readout, shown in a). Bottom: Representative time trace of MCM2-3-Ala. e) NucCDK traces, where MCM2-3-Ala-mNG was used as a nuclear mask, taken every 5 min in EMM media. *n* = 42 cells; 1 of 3 biological repeats. f) NucCDK time traces where the top 15% method was used with representative time trace (same cell as Fig. 1c) shown on the bottom. *n* = 42 cells.

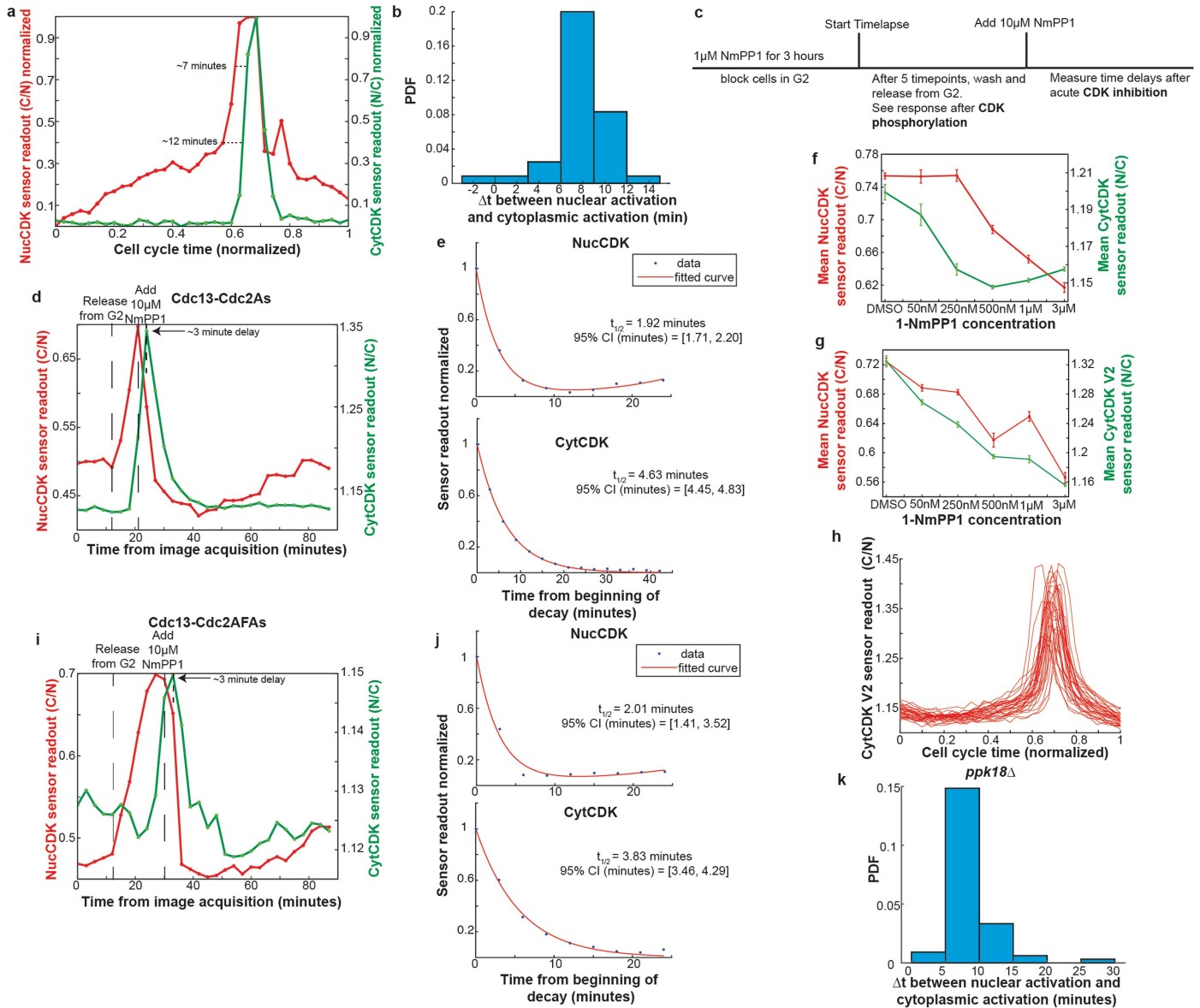

**Extended Data Fig. 2 | Time delay is not due to slow translocation kinetics or differences in sensitivities.** a) Fig. 2c showing time delays at different points. b) Histogram of the delay from nuclear activation to cytoplasmic activation using NucCDK-mScarletI and CytCDK-mNG with 3-min intervals in YE4S. $n = 40$ cells; 1 of 3 biological repeats. c) Schematic of block and release experiments. d) Representative trace in Cdc13-Cdc2As background. 10 μM 1-NmPP1 was added after 8 time points. 3-min delay refers to the delay when readouts begin to drop (shown with dotted lines). e) Decay curves of meaned data after CDK inhibition ($n = 24$ cells; 1 of 2 biological repeats). NucCDK was fit with a two-exponential model to correct for slight rise in readout. Estimated half life with 95% confidence intervals are shown in brackets. f) Mean readouts of sensors in Cdc13-Cdc2AFAs cells released into different concentrations of

1-NmPP1. $n = 185, 49, 57, 144, 218,$ and 108 cells, for DMSO, 50 nM, 250 nM, 500 nM, 1 μM, and 3 μM, respectively; 1 of 2 biological repeats. Error bars represent standard error of the mean. g) same as f except using CytCDK V2. $n = 208, 257, 191, 234, 61,$ and 142 cells for DMSO, 50 nM, 250 nM, 500 nM, 1 μM, and 3 μm, respectively; 1 of 2 biological repeats. h) Representative traces of CytCDK V2. $n = 30$ cells; 1 of 3 biological repeats. i) Representative trace in Cdc13-Cdc2AFAs background. 10 μM 1-NmPP1 was added after 11 time points. j) Decay curves of meaned data after CDK inhibition ($n = 42$ cells; 1 of 2 biological repeats). Estimated half life with 95% confidence intervals are shown in brackets. k) Histogram of the delay from nuclear activation to cytoplasmic activation using the sensors, NucCDK-mScarletI and CytCDK-mNG, in a $ppk18\Delta$ background, using 5 min intervals in YE4S. $n = 66$ cells; 1 of 3 biological repeats.

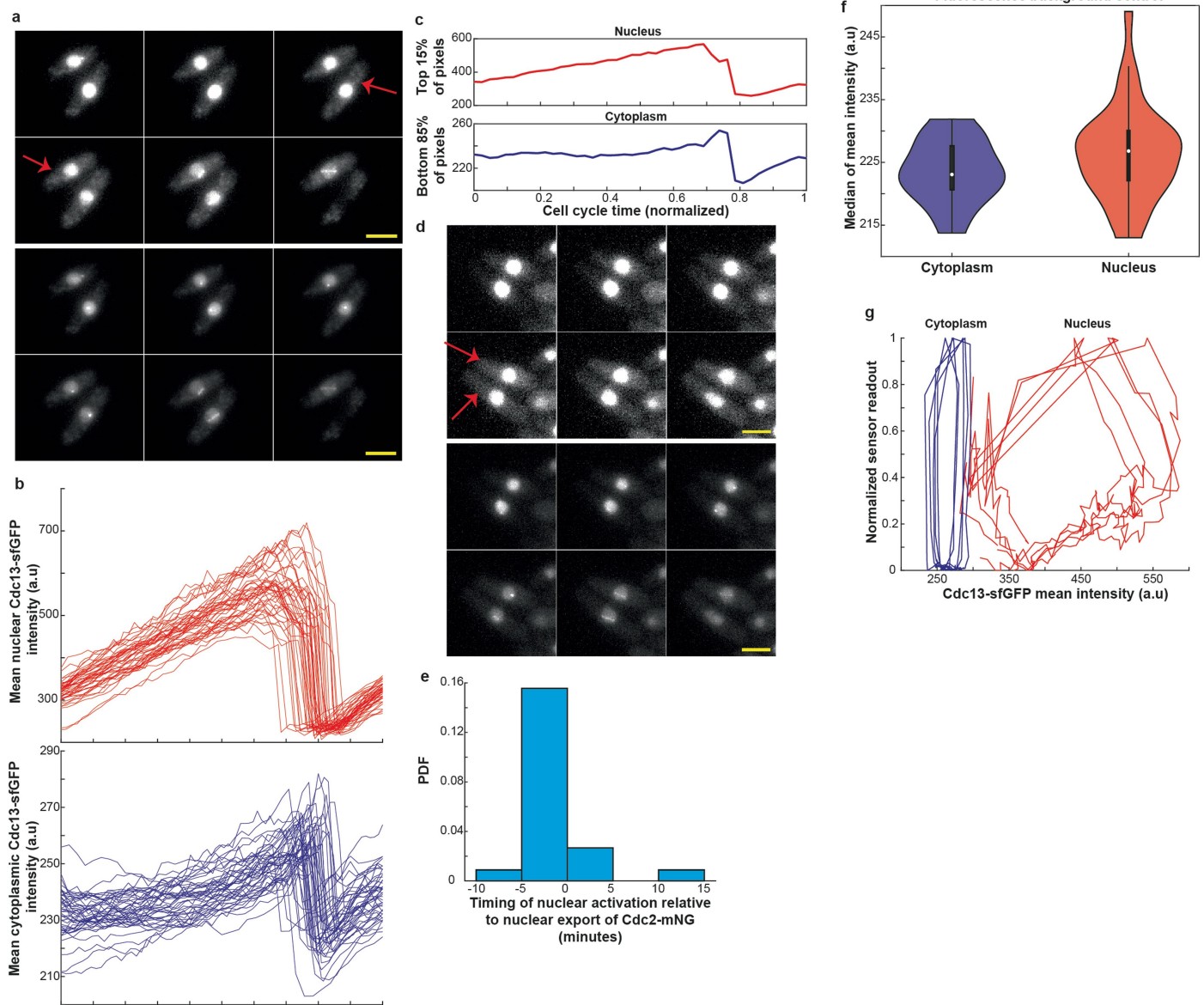

**Extended Data Fig. 3 | Cdc13 and Cdc2 export is associated with cytoplasmic CDK activation.** a) top) Montage of Cdc13-sfGFP showing nuclear export. Representative from 3 biological repeats. Images taken every 5 min in EMM media, and each image represents 5 min. Scale bar = 5 μm. Contrast was adjusted for better visualization of export. Red arrows mark cells where export was visually seen. Bottom) same as top but with no contrast adjustment. b) Single cell traces showing nuclear (top) and cytoplasmic (bottom) mean intensities. MCM2-3-Ala-mScarletI was used as a nuclear mask. *n* = 45 cells; 1 of 4 biological repeats. c) Representative trace of nuclear and cytoplasmic mean intensities using top 15% and bottom 85% of pixels, respectively. Same cell as in Fig. 3a. d) top) Montage of Cdc2-mNG taken every 5 min, in EMM media, showing two cells going through nuclear export. Representative from 3 biological repeats. Each image is 5 min. Scale bar = 5 μm. Contrast was adjusted for better visualization of export. Bottom) Same as top but with no contrast adjustment. e) Histogram of nuclear activation using NucCDK-mScarletI, with respect to Cdc2-mNG nuclear export. Data was collected every 5 min. *n* = 45 cells; 1 of 3 biological repeats. f) Background control of imaging in GFP channel with MCM2-3-Ala-mScarletI under same conditions used in Fig. 3h. White dot represents median value, black solid rectangle represents the IQR (bounded by the 25th and 75th percentiles), and black solid line represents the whiskers bounded by the minimum and maximum defined as data points within 1.5x IQR values. The median value of the mean intensity of the nucleus and cytoplasm over the cell cycle was used. *n* = 33 cells; 1 of 2 biological repeats. g) Phase plots in nucleus and cytoplasm, plotted on the same phase space. Sensor readouts were min max normalized for comparison. 5 cell traces are shown for each cellular compartment.

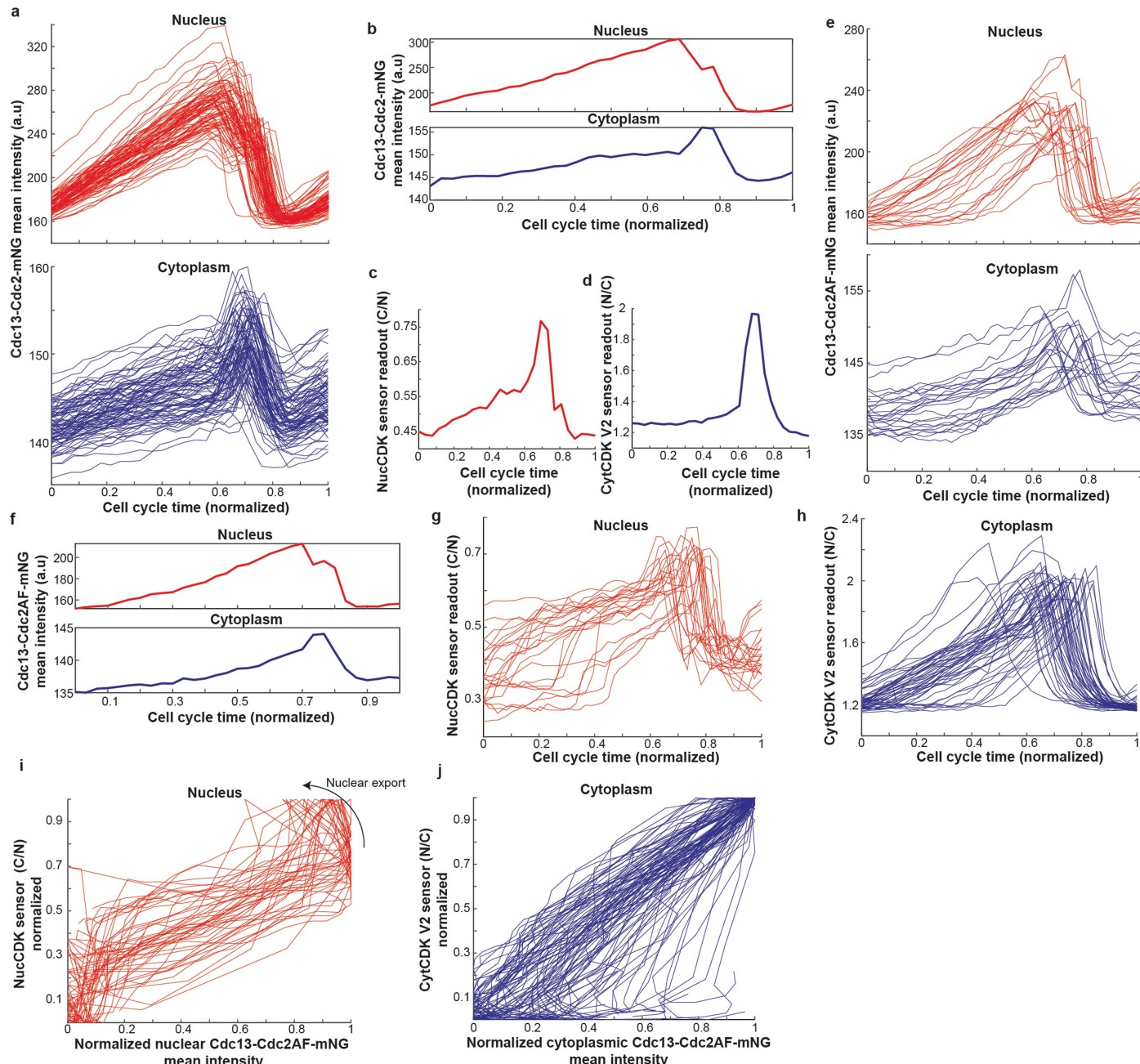

**Extended Data Fig. 4 | Cdc13-Cdc2 fusion protein mimics dynamics of uncomplexed Cdc13.** a) Single-cell traces of Cdc13-Cdc2-mNG mean intensities in nucleus (top) and cytoplasm (bottom). $n$ = 89 cells; 1 of 3 biological repeats. b) Representative trace of Cdc13-Cdc2-mNG mean intensities in the nucleus and cytoplasm. c) Representative trace of NucCDK-mScarletI in Cdc13-Cdc2-mNG background. d) Representative trace of CytCDK V2-mScarletI in Cdc13-Cdc2-mNG background. e) Single-cell traces of Cdc13-Cdc2AF-mNG mean intensities in nucleus (top) and cytoplasm (bottom). $n$ = 24 cells; 1 of 3 biological repeats.

f) Representative trace of Cdc13-Cdc2AF-mNG mean intensities in the nucleus and cytoplasm. g) Single-cell traces of NucCDK sensor readout in Cdc13-Cdc2AF-mNG background. $n$ = 24 cells; 1 of 3 biological repeats. h) Single cell traces of CytCDK V2 sensor readout in Cdc13-Cdc2AF-mNG background. $n$ = 49 cells; 1 of 3 biological repeats. i) Min max normalized phase plots of the nucleus in the Cdc13-Cdc2AF-mNG background. $n$ = 24 cells. j) Min max normalized phase plots of the cytoplasm in the Cdc13-Cdc2AF-mNG background. $n$ = 49 cells.

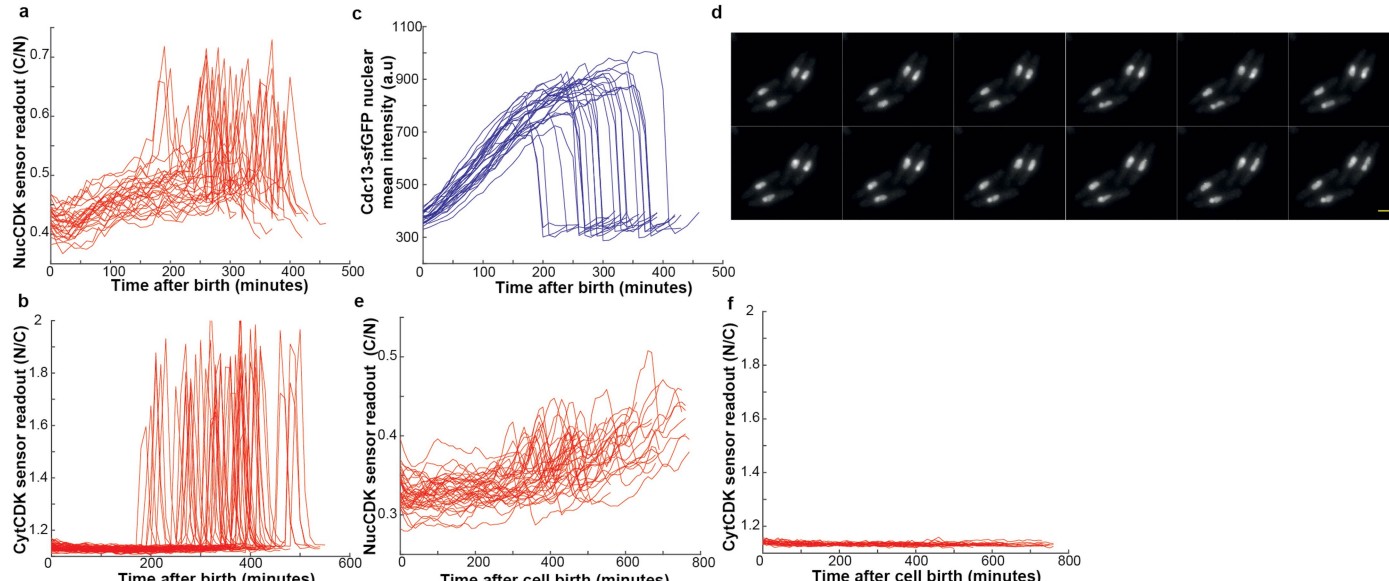

**Extended Data Fig. 5 | Absence of Cdc13 nuclear export and cytoplasmic CDK activation in Cdc13$^{HPM}$ background.** a) Single-cell traces of NucCDK readout in Cdc13WT-sfGFP background, in EMM + thiamine to repress the second copy of Cdc13WT. Images were acquired every 10 min. $n$ = 29 cells; 1 of 3 biological repeats. b) Single-cell traces of CytCDK readout in Cdc13WT-sfGFP in EMM + thiamine to repress the second copy of Cdc13WT. Images were acquired every 10 min. $n$ = 58 cells; 1 of 3 biological repeats. c) Single-cell traces of Cdc13-sfGFP mean nuclear intensity in Cdc13WT-sfGFP background, where the second copy of Cdc13WT was repressed. $n$ = 29 cells. d) Montage of Cdc13HPM-sfGFP. Images taken every 10 min, with each image representing 10 min. Scale bar = 5 µm. Representative of 3 biological repeats. e) Single-cell traces of NucCDK in Cdc13HPM-sfGFP background. $n$ = 36 cells; 1 of 3 biological repeats. f) Traces of CytCDK readout in Cdc13HPM-sfGFP background. Y-axis scaled based on minimum and maximum sensor readout in Cdc13WT -sfGFP background shown in b). $n$ = 23 cells; 1 of 3 biological repeats.

# Reporting Summary

## Statistics

For all statistical analyses, confirm that the following items are present in the figure legend, table legend, main text, or Methods section.

| n/a | Confirmed | |
|-----|-----------|---|
| ☐ | ☒ | The exact sample size (*n*) for each experimental group/condition, given as a discrete number and unit of measurement |
| ☐ | ☒ | A statement on whether measurements were taken from distinct samples or whether the same sample was measured repeatedly |
| ☐ | ☒ | The statistical test(s) used AND whether they are one- or two-sided<br>*Only common tests should be described solely by name; describe more complex techniques in the Methods section.* |
| ☒ | ☐ | A description of all covariates tested |
| ☒ | ☐ | A description of any assumptions or corrections, such as tests of normality and adjustment for multiple comparisons |
| ☐ | ☒ | A full description of the statistical parameters including central tendency (e.g. means) or other basic estimates (e.g. regression coefficient) AND variation (e.g. standard deviation) or associated estimates of uncertainty (e.g. confidence intervals) |
| ☐ | ☒ | For null hypothesis testing, the test statistic (e.g. *F*, *t*, *r*) with confidence intervals, effect sizes, degrees of freedom and *P* value noted<br>*Give P values as exact values whenever suitable.* |
| ☒ | ☐ | For Bayesian analysis, information on the choice of priors and Markov chain Monte Carlo settings |
| ☒ | ☐ | For hierarchical and complex designs, identification of the appropriate level for tests and full reporting of outcomes |
| ☒ | ☐ | Estimates of effect sizes (e.g. Cohen's *d*, Pearson's *r*), indicating how they were calculated |

*Our web collection on statistics for biologists contains articles on many of the points above.*

## Software and code

Policy information about availability of computer code

| Data collection | Microscopy images were collected using MicroManager (uManager, NIH) v2.0.0-beta3 20180704 |
|-----------------|--------------------------------------------------------------------------------------------|
| Data analysis | Image analysis was done partly though FIJI 1.54f (NIH). Image and data analysis was also done using custom scripts with Matlab 2022. All custom scripts can be found at: https://github.com/nkapadia27/Spatiotemporal-Orchestration-of-Mitosis. DOI: 10.5281/zenodo.11072087 |

For manuscripts utilizing custom algorithms or software that are central to the research but not yet described in published literature, software must be made available to editors and reviewers. We strongly encourage code deposition in a community repository (e.g. GitHub). See the Nature Portfolio guidelines for submitting code & software for further information.

## Data

Policy information about availability of data

All manuscripts must include a data availability statement. This statement should provide the following information, where applicable:
- Accession codes, unique identifiers, or web links for publicly available datasets
- A description of any restrictions on data availability
- For clinical datasets or third party data, please ensure that the statement adheres to our policy

All strains, data, and plasmids are  available upon request. Due to the large size of the timelapse microscopy imaging data we have made this data available upon request by contacting the authors.

# Research involving human participants, their data, or biological material

Policy information about studies with [human participants or human data](). See also policy information about [sex, gender (identity/presentation), and sexual orientation]() and [race, ethnicity and racism]().

| | |
|---|---|
| Reporting on sex and gender | Not relevant |
| Reporting on race, ethnicity, or other socially relevant groupings | not relevant |
| Population characteristics | not relevant |
| Recruitment | not relevant |
| Ethics oversight | not relevant |

Note that full information on the approval of the study protocol must also be provided in the manuscript.

# Field-specific reporting

Please select the one below that is the best fit for your research. If you are not sure, read the appropriate sections before making your selection.

☒ Life sciences ☐ Behavioural & social sciences ☐ Ecological, evolutionary & environmental sciences

For a reference copy of the document with all sections, see [nature.com/documents/nr-reporting-summary-flat.pdf]()

# Life sciences study design

All studies must disclose on these points even when the disclosure is negative.

| | |
|---|---|
| Sample size | Sample sizes were not predetermined. For timelapse imaging, we aimed for at least 30-100 cells as our representative sample in our plots. We believe this was an adequate sample size as the phenotype and quantification (including distributions) was consistent across cells and across multiple repeats. For samples that were quantified, clear profiles in the distributions were obtained along with statistical results that gave us confidence in the conclusions we were making using the data. Furthermore, the parameter estimates, distributions, and statistics were consistent across repeats. |
| Data exclusions | Cells that did not properly grow or divide, including ones that led to cell death were excluded from analysis. Traces that were extremely noisy were removed after carefully looking at the raw data, as often these represented issues of image segmentation or poor quality of data. These excluded data often represented a very small fraction (1-5%). |
| Replication | All replicates were successful and we did at least 2 independent replicate experiments. For example, for timelapse imaging, we collected data for the same condition/strain at least on two different days to ensure consistency. Furthermore, we collected multiple fields of view during timelapse imaging. |
| Randomization | Randomization was not relevant to the study, as we only analyzed a single genotype for each experiment. |
| Blinding | Blinding was not relevant, as the phenotype of the cells was obvious when looking on the microscope. The analysis of the phosphoproteomics was based on grouping according to localization so it was not possible to perform this where the investigators were blinded to the groups allocation. |

# Reporting for specific materials, systems and methods

We require information from authors about some types of materials, experimental systems and methods used in many studies. Here, indicate whether each material, system or method listed is relevant to your study. If you are not sure if a list item applies to your research, read the appropriate section before selecting a response.

## Materials & experimental systems

| n/a | Involved in the study |
|-----|----------------------|
| ☒ | Antibodies |
| ☐ | ☒ Eukaryotic cell lines |
| ☒ | Palaeontology and archaeology |
| ☒ | Animals and other organisms |
| ☒ | Clinical data |
| ☒ | Dual use research of concern |
| ☒ | Plants |

## Methods

| n/a | Involved in the study |
|-----|----------------------|
| ☒ | ChIP-seq |
| ☒ | Flow cytometry |
| ☒ | MRI-based neuroimaging |

# Eukaryotic cell lines

Policy information about cell lines and Sex and Gender in Research

| | |
|---|---|
| Cell line source(s) | All S.pombe strains used are mentioned in Supplementary Table 1. The L972 was the background for all strains used in the experiments. |
| Authentication | Authentication was done by PCR and sequencing. |
| Mycoplasma contamination | Not possible |
| Commonly misidentified lines (See ICLAC register) | None |

# Plants

| | |
|---|---|
| Seed stocks | *Report on the source of all seed stocks or other plant material used. If applicable, state the seed stock centre and catalogue number. If plant specimens were collected from the field, describe the collection location, date and sampling procedures.* |
| Novel plant genotypes | *Describe the methods by which all novel plant genotypes were produced. This includes those generated by transgenic approaches, gene editing, chemical/radiation-based mutagenesis and hybridization. For transgenic lines, describe the transformation method, the number of independent lines analyzed and the generation upon which experiments were performed. For gene-edited lines, describe the editor used, the endogenous sequence targeted for editing, the targeting guide RNA sequence (if applicable) and how the editor was applied.* |
| Authentication | *Describe any authentication procedures for each seed stock used or novel genotype generated. Describe any experiments used to assess the effect of a mutation and, where applicable, how potential secondary effects (e.g. second site T-DNA insertions, mosiacism, off-target gene editing) were examined.* |

