## [Peer Review File · Nature]

Spatiotemporal Orchestration of Mitosis by Cyclin-Dependent Kinase

Corresponding Author: Dr Nitin Kapadia

Version 0:

Reviewer comments:

Referee #1

(Remarks to the Author)

This very interesting paper examines the activation of Cdk1 in the nucleus and in the cytoplasm in *S. pombe*. Using kinase translocation reporters of nuclear and cytoplasmic Cdk1 activity, the authors infer that the (1) Cdk1 activation peaks earlier in the cell cycle than previously thought; (2) Cdk1 is activated in nucleus 5-10 min before it is activated in the cytoplasm; (3) the cyclin concentration required to achieve Cdk1 activation is lower in the cytoplasm than in the nucleus; (4) Cdc13-Cdk1 complexes translocate from the nucleus to the cytoplasm immediately after the nuclear Cdk1 is activated; (5) there is hysteresis in the response of nuclear Cdk1 to cyclin, but not in the response of cytoplasmic Cdk1 to cyclin; and the spindle pole body is not required for activation of the nuclear Cdk1, but is required for the appearance of active Cdk1 in the cytoplasm.

Some of these conclusions, like the inferred hysteresis in the nuclear Cdk1 response, fit with the general conceptual picture of Cdk1 regulation that came out of modeling studies and quantitative biochemical studies of *Xenopus* extracts. Some, like the shift of Cdc13-Cdk1 out of the nucleus at the onset of mitosis, are pretty much opposite to what happens in mammalian cell lines, where cyclin B1 translocates *into* the nucleus just prior to nuclear envelope breakdown. And some, like the SPB role, turn previous *pombe* results on their collective head.

The paper is clearly written, the data are easy to understand, the subject matter is of general interest, and the conclusions are very interesting. However, I have two main reservations about the work as it stands. If these two major points can be cleared up, I would feel much more confident that the paper's main conclusions are correct.

Major points:

1. The process that links a change in Cdk1 activity to a change in sensor localization consists of at least two steps, the phosphorylation and then the translocation. Each step will introduce a time lag into the process, and I don't have a sense for whether this will be negligible or substantial. This is important to know. If the time lag is substantial, then maybe Cdk1 peaks even earlier in the cell cycle than the present data imply, and if the time lags are substantial and different for the nuclear vs. cytoplasmic reporters, then the conclusion that nuclear activation precedes cytoplasmic activation would be in doubt.

Can you acutely perturb Cdk1 activity up and down and see how long it takes for the two sensors to respond? I would think that a conventional small molecule Cdk1 inhibitor, or the Shokat analog in a strain expressing AS-Cdk1, should work for this: add the drug to turn Cdk1 activity down, wash it out to turn Cdk1 up.

2. The concentration of cyclin required to half-maximally activate the two sensors will depend upon on how good the sensors are as substrates of Cdk1 and the opposing phosphatase(s). There is no guarantee that the two sensors, one based on Cut3 and one based on Mcm2-3, will have the same EC50, so it is not certain that the cyclin threshold for cytoplasmic activation of Cdk1 is actually lower than it is in the nucleus.

Is there a way to check whether the EC50's of the two sensors are the same, or close to the same?

Minor points:

3. The lettering in the figures is too small.

4. Please test the statistical significance of the earlier phosphorylation of nuclear substrates relative to the other classes of substrates in the re-analysis of the Swaffer et al. data (Fig 1a).

5. A reference to earlier work on translocation-based sensors (e.g. Spencer and Meyer 2013 or Regot and Covert 2014) would be good to add.

6. I am glad to see the data in Fig 4, but I don't think the interpretation is quite right. Hysteresis—a steady state property of a circuit with a built-in bistable switch—can allow an oscillator to have a big fat orbit (limit cycle) in the phase plane, but it is not necessary. One can have a big fat orbit without there being hysteresis in the system's steady state responses, and without the oscillator having a positive feedback trigger. So the wording on lines 187, 214, 218 isn't quite right.

That said, the fact that the right-hand side of the orbit is almost vertical does mean that the fission yeast oscillator probably includes a fast bistable trigger, as long suspected. That should be pointed out.

7. Whenever single trajectories are shown (e.g. not Fig 4a bottom, 4b, 4c, 5f, but Fig 1a, c, d, and most of the other figure panels) it would be helpful to plot the data points as well as the lines connecting the data points. This would be especially helpful for Fig 4a top, as it would give the reader an idea of how fast this relaxation oscillator does its upstroke (nuclear activation/export) versus its build up (cyclin accumulation) and die out (cyclin degradation) phases.

8. Line 304. I'm not so sure that the conclusion of Ref 58 (Yao et al. NCB 2008)—namely that the Rb-E2F circuit is a bistable switch— is correct. See Cornwell et al. Nature 2023 for what to me seems like convincing evidence to the contrary.

Referee #2

(Remarks to the Author)

This paper addresses the mechanisms controlling the initiation of mitosis in eukaryotic cells. The key regulator of this process is Cdk1 in a complex with mitotic cyclin (primarily cyclin B in animals or Cdc13 in the fission yeast model system used here). Cdk1 activation in mitosis depends on removal of inhibitory phosphate(s), and interactions among the inhibitory kinase (Wee1) and activating phosphatase (Cdc25) that control this modification generate positive feedback, resulting in switch-like Cdk1 activation in early mitosis.

Past studies in vertebrate cells provide clues about the subcellular location of the initial site of Cdk1 activation. In these cells, cyclin B is cytoplasmic in interphase but in late prophase rushes into the nucleus, where it triggers nuclear envelope breakdown. Early studies of cyclin B1 phosphorylation (Jackman & Pines 2003) suggested that Cdk1-cyclin B seems to be activated first at the centrosome in prophase, leading in the 2000s to the view that the centrosome is the site of Cdk1 activation. This view has not been well supported by more recent analyses. For example, Santos et al (2012) provided evidence that the positive feedback that drives Cdk1 activation is triggered by its nuclear import. More recently, a pair of important papers from the Gelens and Ferrell labs (Nolet et al, Afanjar et al, 2020) showed that waves of Cdk1 activation in mitotic *Xenopus* extracts are initiated at the nucleus and not the centrosome.

In the current paper, the authors perform an elegant series of experiments supporting the idea that the nucleus is the initial site of Cdk1 activation in fission yeast. They provide evidence in Fig. 1 that the earliest mitotic Cdk1 substrates are nuclear proteins, and in Fig 2 they use Cdk1 activity biosensors to show that nuclear Cdk1 activity rises abruptly just before cytoplasmic activity. Figure 3 then addresses the localization of Cdc13 during the cell cycle, revealing that this cyclin is nuclear throughout the cycle and drops slightly in mitosis, when some portion of the Cdc13-Cdk1 complex is exported to the cytoplasm.

Figure 4 provides phase plots of nuclear cyclin concentration vs Cdk1 activity during mitosis, much like the phase plots that have been explored in depth in *Xenopus* extracts for many years. These plots provide a convenient graphic illustration of the bistability of Cdk1 activity: i.e., Cdk1 kinase activity per cyclin is initially low as cyclin levels rise in mitosis, but then rises abruptly at high cyclin levels due to positive feedback. On the return (leftward) journey during mitotic exit, Cdk1 activity is now higher per cyclin as the cyclin molecules are degraded. As expected, we see in Fig. 4 that the bistable behavior of Cdk1 activity in the fission yeast nucleus depends on inhibitory phosphorylation.

Fig. 4 also shows similar results with cytoplasmic Cdk1-Cdc13, but within the much narrower range of cyclin concentrations seen in Fig 3a. There is less evidence of hysteresis here, presumably because these data reflect dephosphorylated Cdk1-Cdc13 that was exported after its activation in the nucleus.

Finally, in Fig 5 the authors study a cyclin mutant that does not associate with the SPB to show that nuclear Cdk1 activation does not require SPB association – contrary to the view that the SPB is the site of initial Cdk1 activation.

In summary, the results provide quite compelling evidence that Cdk1 activation by the cyclin Cdc13 in fission yeast begins inside the nucleus. This paper therefore provides a thorough analysis *in vivo* to support previous evidence, mentioned above, that the nucleus is the origin of Cdk1 activation in vertebrates.

The authors emphasize multiple times that their work goes against the 'prevailing' or 'predominant' view that Cdk1 activation begins at the centrosome or SPB. I'm not sure that in 2024 this is actually the prevailing view among the experts, although it might remain a common belief in a general reader. The current results are therefore not paradigm-shifting but will help drive home the point that the nucleus is the critical site of Cdk1 activation.

There is also a tendency here to downplay results in *Xenopus* extracts as 'in vitro', implying that they are not necessarily reflective of the true 'in vivo' situation. This may be true in the strictest sense, but I don't think there is much doubt in the field that undiluted *Xenopus* cytoplasm has proven to be an excellent representation of the *in vivo* condition.

Other points to consider:

1. The phosphorylation state of any protein depends in part on the local activity of phosphatases acting on that protein. The authors should consider mentioning that their proteomic analyses and biosensors are not simply a reflection of Cdk1 activity but might also be influenced by phosphatases, which are well known to undergo major activity changes during mitosis. It is conceivable, for example, that the relative timing of nuclear and cytoplasmic sensors is partly the result of differences in phosphatase activities in the two locations.
2. In Fig 1a, are the nuclear envelope and SPB proteins on the inner or outer face of the nuclear envelope?
3. In Fig 2c, why does nuclear Cdk1 sensor continue to be phosphorylated long after the cytoplasmic sensor?
4. In Fig 2f, the cytoplasmic sensor V2 seems to remain phosphorylated longer than V1 in Fig. 2c. Is this reproducible? Is this sensor less sensitive to phosphatases, and therefore a less effective readout of Cdk1 inactivation timing?
5. None of the cell cycle time courses are marked with landmark events, which would be helpful for a general reader. The text often mentions these events (SPB separation, etc.) but they are not indicated.

Version 1:

Reviewer comments:

Referee #1

(Remarks to the Author)

The authors have improved the paper in some small ways, but the two major limitations of the paper remain and have not been mitigated. The first is the issue of how much of a time lag there is between changes in Cdk1 activity and changes in the probes' translocation, and whether the time lags for the two probes are the same or different. The authors point out that the time lag could be as little as a few seconds, and that is true. Or it could be 7 min, as Spencer et al. found for their translocation probe in mammalian cells (Spencer et al. Cell 2013), or.... You can't tell without making the measurement. Measuring a translocation reporter's response to an inhibitor is a basic essential control experiment, both for assessing the specificity of the reporter (see, for example, Regot et al. Cell 2014 and Schwarz et al. Mol Cell 2018) and, in the present case, for determining the dynamics of the probe's response. In the rebuttal the authors mention that the CellAsics plates they used are not compatible with use of the Nmpp1 inhibitor, but surely there are other ways to carry out the experiment. For example, Hauf's lab used lectin-coated Ibidi dishes to immobilize *S. pombe* for fluorescence microscopy (Kamenz and Hauf Curr Biol 2014). The authors also point out that they do not have experience with other small molecule Cdk inhibitors, but why not try them? And finally, the authors point out that the re-examined proteomics data are consistent with late activation of cytoplasmic Cdk1. This is true, but it is the translocation reporter data that most directly makes the point that it is the timing of nuclear vs. cytoplasmic Cdk1 activation rather than differences in the concentrations of nuclear vs. cytoplasmic Cdk1-Cdc13 and/or the properties of the nuclear vs. cytoplasmic substrates that makes the nuclear substrate phosphorylations occur earlier than the cytoplasmic substrate phosphorylations. The authors need to do the control experiments; without the controls, the translocation reporter experiments are not sufficiently definitive.

The second is the issue of whether the two translocation probes, one based on Cut3 and one based on Mcm2-3, are equally sensitive to Cdk activation. The authors point out that they have already made two different probes (CytCDK and CytCDK V2) with different sensitivities, and yet these probes do not seem to differ in terms of their rate change points. This is true, but it is also true that the properties of a kinase substrate can influence the timing of its phosphorylation, even if it doesn't seem to for these two particular substrates. The Nurse lab's own previous work (e.g. Swaffer et al. Cell 2016) supports the importance of the properties of the substrate, and extensive work in budding yeast from Morgan, Loog, Uhlmann and other does as well. Given that the present paper's most important conclusions and most direct experimental approaches rely on the implicit assumption that the two translocation reporters respond rapidly and essentially identically to changes in the local Cdk1 activity, this assumption needs to be tested more thoroughly.

I still find the data interesting and the conclusions of the paper plausible, but without what I consider to be essential control experiments I do not support publication.

Referee #2

(Remarks to the Author)

I thank the authors for providing a thorough and compelling response to my comments. I have no further concerns.

Version 2:

Reviewer comments:

Referee #1

(Remarks to the Author)

The new control experiments solidify the authors' interpretations. I recommend publication. Nice work!

Referees' comments:

Referee #1 (Remarks to the Author):

This very interesting paper examines the activation of Cdk1 in the nucleus and in the cytoplasm in *S. pombe*. Using kinase translocation reporters of nuclear and cytoplasmic Cdk1 activity, the authors infer that the (1) Cdk1 activation peaks earlier in the cell cycle than previously thought; (2) Cdk1 is activated in nucleus 5-10 min before it is activated in the cytoplasm; (3) the cyclin concentration required to achieve Cdk1 activation is lower in the cytoplasm than in the nucleus; (4) Cdc13-Cdk1 complexes translocate from the nucleus to the cytoplasm immediately after the nuclear Cdk1 is activated; (5) there is hysteresis in the response of nuclear Cdk1 to cyclin, but not in the response of cytoplasmic Cdk1 to cyclin; and the spindle pole body is not required for activation of the nuclear Cdk1, but is required for the appearance of active Cdk1 in the cytoplasm.

Some of these conclusions, like the inferred hysteresis in the nuclear Cdk1 response, fit with the general conceptual picture of Cdk1 regulation that came out of modeling studies and quantitative biochemical studies of *Xenopus* extracts. Some, like the shift of Cdc13-Cdk1 out of the nucleus at the onset of mitosis, are pretty much opposite to what happens in mammalian cell lines, where cyclin B1 translocates *into* the nucleus just prior to nuclear envelope breakdown. And some, like the SPB role, turn previous *pombe* results on their collective head.

The paper is clearly written, the data are easy to understand, the subject matter is of general interest, and the conclusions are very interesting. However, I have two main reservations about the work as it stands. If these two major points can be cleared up, I would feel much more confident that the paper's main conclusions are correct.

The positive comments of the Reviewer are much appreciated. The two major points they raised are dealt with by our responses in the next two sections.

Major points:

1. The process that links a change in Cdk1 activity to a change in sensor localization consists of at least two steps, the phosphorylation and then the translocation. Each step will introduce a time lag into the process, and I don't have a sense for whether this will be negligible or substantial. This is important to know. If the time lag is substantial, then maybe Cdk1 peaks even earlier in the cell cycle than the present data imply, and if the time lags are substantial and different for the nuclear vs. cytoplasmic reporters, then the conclusion that nuclear activation precedes cytoplasmic activation would be in doubt.

Can you acutely perturb Cdk1 activity up and down and see how long it takes for the two sensors to respond? I would think that a conventional small molecule Cdk1 inhibitor, or the Shokat analog in a strain expressing AS-Cdk1, should work for this: add the drug to turn Cdk1 activity down, wash it out to turn Cdk1 up.

We agree with the Reviewer that sensor localization changes involve steps that can influence the measurements of time lags in CDK activation in the nucleus and cytoplasm. The major conclusion we make with this experiment is that CDK in the nucleus is activated prior to CDK in the cytoplasm being activated, as indicated by our sensor experiment (for example in Figure 2). This conclusion is supported by the completely different experimental approach of phosphoproteomics (Figure 1A, and extended data Figure 1a) which shows that late substrates in the nucleus become phosphorylated before those in the cytoplasm. The large majority of these substrates are not translocating after CDK phosphorylation, but we still observe a significant delay in the phosphorylation of cytoplasmic substrates compared to nuclear substrates. This result is consistent with CDK activation in the cytoplasm occurring after CDK activation in the nucleus. We are not aware of a systematic study of nuclear import/export rates in fission yeast, but in budding yeast it has been reported using fluorescence recovery after photobleaching (FRAP), that the timescale of import/export of proteins is a few seconds (Lucia Durrieu et al. 2023, iScience), which if similar in fission yeast, is short compared with the time delay we measure with the sensors, so it is unlikely to introduce a significant time lag. Our sensors do not contain any binding motifs that could cause them to associate with other proteins or cellular structures that would slow down their import/export. However, the Reviewer is right to comment on the effects the time lags due to translocation, and we now refer to this in the revised manuscript (see lines 118-120)

The Cdk1 perturbation experiment suggestion made by the Reviewer is an elegant proposal and we gave some thought as to how we might do such an experiment. We use a CellAsics microfluidics to rapidly change media over the course of fluorescence timelapse imaging, but unfortunately, Nmpp1 (used to inhibit Cdk1-As strains), is absorbed into the CellAsics plates and so cannot be used in this experiment. We do not have any experience with small Cdk1 inhibitors in fission yeast and differing rates of incorporation of the Nmpp1 inhibitor into different cellular compartments may influence estimation of time lags. However, the independent phosphoproteomics data discussed above is consistent with the order of activation between the nucleus and cytoplasm confirming the main point we are making with this experiment.

2. The concentration of cyclin required to half-maximally activate the two sensors will depend upon on how good the sensors are as substrates of Cdk1 and the opposing phosphatase(s). There is no guarantee that the two sensors, one based on Cut3 and one based on Mcm2-3, will have the same EC50, so it is not certain that the cyclin threshold for cytoplasmic activation of Cdk1 is actually lower than it is in the nucleus.

Is there a way to check whether the EC50's of the two sensors are the same, or close to the same?

We also agree with the Reviewer that care should be taken in interpreting sensor experiments, as they can be influenced by the sensitivity of sensors to Cdk1 phosphorylation. In our experiments investigating Cdc13 thresholds in the nucleus and cytoplasm for CDK activation, we did not measure EC50 but rather the “rate change point” in sensor readout that linked the Cdc13 thresholds to points of CDK activation in

the nucleus and cytoplasm. To test if the rate change point changes with sensors of different sensitivities we used two different cytoplasmic CDK sensors (CytCDK and CytCDK V2, Fig 2e and f). The details of how CytCDK V2 was constructed to have increased sensitivity are explained in the Methods under “Development of NucCDK and CytCDK sensors”. Despite the difference in sensitivities, the rate change point was the same for CytCDK and CytCDK V2 (Figure 2). Therefore, given the rate change point doesn’t change, our measurements for Cdc13 thresholds are unlikely to be significantly influenced by the sensitivity of the sensors used. Also, the concentration of Cdc13 in the nucleus and cytoplasm are clearly different at these rate change points. Given the clear differences in concentrations, we think it is safe to conclude that the cytoplasm and nuclear CDK’s require different cyclin levels. However, we bring this issue that the Reviewer has raised to the attention of the reader (lines 162-163). The differences in the shapes of the phase plots are less influenced by the sensitivity issue (Extended Data Fig. 3g, Figure 4a), because the intrinsic sensitivity of the substrate as dictated by its amino acid sequence doesn’t change over the cell cycle (i.e. if the substrate is sensitive to net CDK activity during interphase, it will also be sensitive in mitosis).

Minor points:

3. The lettering in the figures is too small.

We have increased the size of the lettering in the figures within Nature’s guidelines on font size.

4. Please test the statistical significance of the earlier phosphorylation of nuclear substrates relative to the other classes of substrates in the re-analysis of the Swaffer et

al. data (Fig 1a).

We have done a statistical significance test of the earlier nuclear substrate phosphorylation compared to the other compartments and it is shown in Figure 1A. We used our rate change algorithm to identify points of changes in substrate phosphorylation due to Cdk1 activation at mitotic onset, as substrates within a spatial compartment can vary slightly in their timing of initial phosphorylation due to their sensitivity. We used an unequal, two-sample, left-tailed t-test, comparing nuclear substrates with each of the other compartments. The tests indicate that the differences of nuclear substrates compared to substrates in other compartments are statistically significant.

5. A reference to earlier work on translocation-based sensors (e.g. Spencer and Meyer 2013 or Regot and Covert 2014) would be good to add.

We agree. We have included the reference to Spencer et al (2013) in the Methods section when describing the development of the NucCDK sensor (lines 555-557)

6. I am glad to see the data in Fig 4, but I don't think the interpretation is quite right. Hysteresis—a steady state property of a circuit with a built-in bistable switch—can allow an oscillator to have a big fat orbit (limit cycle) in the phase plane, but it is not necessary. One can have a big fat orbit without there being hysteresis in the system's steady state responses, and without the oscillator having a positive feedback trigger. So the wording on lines 187, 214, 218 isn't quite right.

We thank the reviewer for pointing this out. In the original submitted manuscript, we tried to distinguish the difference between dynamic oscillations and the steady state hysteresis response at the beginning of the Results section on bistability, but now realize we may have overstated our claims given that we are not looking at steady-state

measurements. We agree that oscillations can be achieved without hysteresis as shown theoretically in Pomerening et al (2005), where fast rates of cyclin synthesis and degradation can result in oscillations without bistability. We have added lines 173-174 to mention how oscillations can be achieved even without hysteresis. Furthermore, following the Reviewer's comments, we have toned down our claims of showing hysteresis in the Results section, removing sentences that make claims that could be considered too strong, and now stating that our results are "likely" to be indicative of an underlying steady-state hysteretic loop (see lines 223-225).

However, we are of the view that the CDKY15 feedback loops are largely responsible for the stable oscillations as the CDKAF collapse the orbits and makes the responses approximately linear (highlighted now in lines 216-217). This was shown theoretically in Pomerening et al (2005), where the phase response becomes increasingly linear as the feedback factor decreases even when cyclin synthesis and degradation are fast. The CDKAF collapses the orbits in a similar way to the CDKAF dynamic plot shown in Pomerening et al (2005) using *Xenopus* extracts, which also collapses into a narrow loop. These authors proposed that this is in part due to the collapse of the steady-state hysteresis loop. Therefore, while we cannot definitively say there is hysteresis (as we did not measure the steady-state responses), the evidence overall suggests that there is a hysteretic circuit due to CDKY15 feedback loops.

That said, the fact that the right-hand side of the orbit is almost vertical does mean that the fission yeast oscillator probably includes a fast bistable trigger, as long suspected. That should be pointed out.

We thank the Reviewer for this comment and have now mentioned this in lines 190-191.

7. Whenever single trajectories are shown (e.g. not Fig 4a bottom, 4b, 4c, 5f, but Fig 1a,

c, d, and most of the other figure panels) it would be helpful to plot the data points as well as the lines connecting the data points. This would be especially helpful for Fig 4a top, as it would give the reader an idea of how fast this relaxation oscillator does its upstroke (nuclear activation/export) versus its build up (cyclin accumulation) and die out (cyclin degradation) phases.

We agree with the suggestion and that it would be informative to understand the temporal dynamics of the oscillator, especially with regards to data in Figure 4. We have now plotted the data points in Figure 4a,d, and e, marked with green circles. We plotted the data points for the figure panels but found it made the figures overly crowded and made it difficult to easily discern the points of interest that we wished to stress within a given figure (e.g. with Figure 1a, plotting the data points made it difficult to discern the differences in substrate phosphorylation). Therefore, we thought it would be better to not plot the data points for the other figures as the properties of the oscillator were not the focus of the other figures, but we agree it is informative for data in Figure 4 which we have now included.

8. Line 304. I'm not so sure that the conclusion of Ref 58 (Yao et al. NCB 2008)—namely that the Rb-E2F circuit is a bistable switch— is correct. See Cornwell et al. Nature 2023 for what to me seems like convincing evidence to the contrary.

We agree and have removed the reference in lieu of the recent findings by Cornwell et al Nature 2023. Instead, we now have included a reference to Verdugo et al (2013) which highlights how bistability with recurring motifs might be common across different cell transitions.

Referee #2 (Remarks to the Author):

This paper addresses the mechanisms controlling the initiation of mitosis in eukaryotic cells. The key regulator of this process is Cdk1 in a complex with mitotic cyclin (primarily cyclin B in animals or Cdc13 in the fission yeast model system used here). Cdk1 activation in mitosis depends on removal of inhibitory phosphate(s), and interactions among the inhibitory kinase (Wee1) and activating phosphatase (Cdc25) that control this modification generate positive feedback, resulting in switch-like Cdk1 activation in early mitosis.

Past studies in vertebrate cells provide clues about the subcellular location of the initial site of Cdk1 activation. In these cells, cyclin B is cytoplasmic in interphase but in late prophase rushes into the nucleus, where it triggers nuclear envelope breakdown. Early studies of cyclin B1 phosphorylation (Jackman & Pines 2003) suggested that Cdk1-cyclin B seems to be activated first at the centrosome in prophase, leading in the 2000s to the view that the centrosome is the site of Cdk1 activation. This view has not been well supported by more recent analyses. For example, Santos et al (2012) provided evidence that the positive feedback that drives Cdk1 activation is triggered by its nuclear import. More recently, a pair of important papers from the Gelens and Ferrell labs (Nolet et al, Afanjar et al, 2020) showed that waves of Cdk1 activation in mitotic *Xenopus* extracts are initiated at the nucleus and not the centrosome.

In the current paper, the authors perform an elegant series of experiments supporting the idea that the nucleus is the initial site of Cdk1 activation in fission yeast. They provide evidence in Fig. 1 that the earliest mitotic Cdk1 substrates are nuclear proteins,

and in Fig 2 they use Cdk1 activity biosensors to show that nuclear Cdk1 activity rises abruptly just before cytoplasmic activity. Figure 3 then addresses the localization of Cdc13 during the cell cycle, revealing that this cyclin is nuclear throughout the cycle and drops slightly in mitosis, when some portion of the Cdc13-Cdk1 complex is exported to the cytoplasm.

Figure 4 provides phase plots of nuclear cyclin concentration vs Cdk1 activity during mitosis, much like the phase plots that have been explored in depth in *Xenopus* extracts for many years. These plots provide a convenient graphic illustration of the bistability of Cdk1 activity: i.e., Cdk1 kinase activity per cyclin is initially low as cyclin levels rise in mitosis, but then rises abruptly at high cyclin levels due to positive feedback. On the return (leftward) journey during mitotic exit, Cdk1 activity is now higher per cyclin as the cyclin molecules are degraded. As expected, we see in Fig. 4 that the bistable behavior of Cdk1 activity in the fission yeast nucleus depends on inhibitory phosphorylation.

Fig. 4 also shows similar results with cytoplasmic Cdk1-Cdc13, but within the much narrower range of cyclin concentrations seen in Fig 3a. There is less evidence of hysteresis here, presumably because these data reflect dephosphorylated Cdk1-Cdc13 that was exported after its activation in the nucleus.

Finally, in Fig 5 the authors study a cyclin mutant that does not associate with the SPB to show that nuclear Cdk1 activation does not require SPB association – contrary to the view that the SPB is the site of initial Cdk1 activation.

In summary, the results provide quite compelling evidence that Cdk1 activation by the cyclin Cdc13 in fission yeast begins inside the nucleus. This paper therefore provides a thorough analysis in vivo to support previous evidence, mentioned above, that the nucleus is the origin of Cdk1 activation in vertebrates.

The authors emphasize multiple times that their work goes against the ‘prevailing’ or ‘predominant’ view that Cdk1 activation begins at the centrosome or SPB. I’m not sure that in 2024 this is actually the prevailing view among the experts, although it might remain a common belief in a general reader. The current results are therefore not paradigm-shifting but will help drive home the point that the nucleus is the critical site of Cdk1 activation.

The positive comments of the Reviewer are much appreciated. On rereading our manuscript we agree we have mentioned too much that our work goes against the prevailing model and have modified our manuscript in several ways to deal with this (see for example, lines 10,59,230-231, and 276-277).

We acknowledge the important *Xenopus* extract studies mentioned by the Reviewer, which suggested that the nucleus acts as the “pacemaker”, in lines 62-63, 325-326. These *Xenopus* extracts studies used markers of mitosis such as nuclear envelope breakdown and/or disassembly of microtubule arrays to find the origins of mitotic waves but did not investigate the spatial dynamics of Cdk1 activity directly, so believe our work is of interest in this regard. Also, we have discussed the Santos et al (2012) paper that the Reviewer cites, in lines 139-140, 329-330, 344-345, though this paper mainly examined the role of the nucleus in facilitating nuclear accumulation of cyclin B1-Cdk1, and not the spatial origin of Cdk1 activation. This is demonstrated in the

mathematical modelling described in Santos et al (2012) which still assumes that Cdk1 is initially activated in the cytoplasm.

The spatial origin of Cdk1 activation within the nucleus in living cells was one of the main findings in our manuscript, but in addition, we have also contributed to understanding other important aspects of CDK control over mitosis, including CDK nuclear and cytoplasmic activation domains, translocation between the nucleus and cytoplasm, bistability/hysteresis, linking CDK control with DNA status, and the centrosome as a signal relay. These emphasize the spatiotemporal aspects *in vivo* how CDK brings about mitosis in a coordinated manner throughout the cell.

There is also a tendency here to downplay results in *Xenopus* extracts as 'in vitro', implying that they are not necessarily reflective of the true 'in vivo' situation. This may be true in the strictest sense, but I don't think there is much doubt in the field that undiluted *Xenopus* cytoplasm has proven to be an excellent representation of the *in vivo* condition.

We certainly did not mean to imply that *Xenopus* extracts are not a good model. They absolutely are a good model. We also do not think calling it *in vitro* downplays the model. Many *Xenopus* researchers also call it *in vitro* including in the paper that the Reviewer cites (Nolet et al (2020)). We only wanted to emphasize our work has been done in natural intact cells. However, to avoid any misunderstanding we changed our manuscript from *in vitro* and refer now to these experimental systems as *Xenopus* extracts (see lines 62-63, 170, 325).

Other points to consider:

1. The phosphorylation state of any protein depends in part on the local activity of phosphatases acting on that protein. The authors should consider mentioning that their proteomic analyses and biosensors are not simply a reflection of Cdk1 activity but might also be influenced by phosphatases, which are well known to undergo major activity changes during mitosis. It is conceivable, for example, that the relative timing of nuclear and cytoplasmic sensors is partly the result of differences in phosphatase activities in the two locations.

The Reviewer makes an important point on the role of phosphatases. Phosphatases are reduced in activity at the onset of mitosis and have been implicated in increasing substrate phosphorylation at mitotic onset and have been proposed to be responsible for hysteresis in substrate phosphorylation, including in *Xenopus* extracts (Grallet.A, et al, Nature (2015), Kamenz.J, et al, Current Biology (2021)). Cdk1 activity and PP2A/B55 (considered to be the main antagonistic phosphatase of Cdk1 substrate phosphorylation) are linked through the Greatwall Kinase. Therefore, to consider the point raised by the Reviewer, we have carried out a further experiment investigating whether the activity of Greatwall may influence the timing of inactivation of phosphatases and the spatial differences in substrate phosphorylation and the time delays observed with our sensors. We used a strain deleted for the gene encoding the main Greatwall kinase in fission yeast (*ppk18*) and it had no significant effect on the delay in the rate change points of our sensors, NucCDK and CytCDK (see below). We also did not notice a significant change in the overall pattern of the sensor readouts.

We have now included this new data as Extended Data Figure 2c and have included lines 125-132 to address the Greatwall-endosulfine-PP2A phosphatase pathway.

The fact that the CDKAF essentially linearizes the sensor readouts and collapses the phase plots, also suggests that CDK15 feedback loops are major drivers responsible for the rapid rise in substrate phosphorylation (and the rate changes in sensor readouts) at mitotic onset and the hysteretic responses, in both the nucleus and cytoplasm. We have now added lines 225-227 highlighting the AF data in the context of substrate phosphorylation.

For the attention of the Reviewer, our lab is currently investigating the role of phosphatases in the fission yeast cell cycle. Another lab member has degraded Pab1 (B55) to see effects on CytCDK, and have found that the rate change point in sensor readout was identical to CytCDK in wild type cells. This work is still in progress but will be submitted for publication soon.

2. In Fig 1a, are the nuclear envelope and SPB proteins on the inner or outer face of the nuclear envelope?

We do not know which or what fraction of the nuclear envelope proteins are localized on the outer face. The SPB is on the outer face of the nuclear envelope in G2 and inserts into the nuclear membrane around the time when the mitotic spindle begins to form. We found that spindle formation happens after CDK activation in the nucleus, and therefore at the time of nuclear CDK activation, we assume that the SPB is likely to be facing the cytoplasm.

3. In Fig 2c, why does nuclear Cdk1 sensor continue to be phosphorylated long after the cytoplasmic sensor?

We think the NucCDK continues to be phosphorylated after the cytoplasmic sensor CytCDK because it is likely a more sensitive sensor to net CDK activity (CDK minus phosphatase), compared to the cytoplasmic sensor, and thus can detect activity at lower levels of CDK activity compared to CytCDK. In addition, as Figure 2F illustrates, we have addressed the sensitivity issue by making an altered version of CytCDK by mutating T19 to serine, CytCDK V2, that has increased sensitivity to net CDK activity. Serines are better phosphorylated by CDK while also becoming worse at being dephosphorylated by PP2A (see Methods section). Both these factors contribute to making CytCDK V2 a more sensitive sensor of net CDK activity.

4. In Fig 2f, the cytoplasmic sensor V2 seems to remain phosphorylated longer than V1 in Fig. 2c. Is this reproducible? Is this sensor less sensitive to phosphatases, and therefore a less effective readout of Cdk1 inactivation timing?

As mentioned above, the V2 sensor is probably less sensitive to phosphatases and also more sensitive to CDK activity. It may influence the interpretation of Cdk1 inactivation timing but we think it is more likely to influence the rate of inactivation, rather than the actual timing of when inactivation begins (see also Extended Data Figure 2b). The experiments in Figure 2 were done in a wild type background with other cyclins present including G1 and S phase cyclins. Fission yeast cells have a short G1 and normally proceed through G1/S prior to physical separation of the daughter cells. It is possible that other cyclin-Cdc2 complexes rather than Cdc13-Cdc2 contribute to the elevated phosphorylation with CytCDK V2 after one would expect Cdc13 to be degraded. In support of this, when we used the Cdc13-Cdc2 fusion protein where other cyclins are unlikely to contribute to CDK activity, we see a less pronounced rate of inactivation (compare Figure 2f and Extended Data 4d). We've also included a representative trace here for the Reviewer showing the mean cytoplasmic Cdc13-Cdc2-mNG intensity and CytCDK V2 readout.

It should be noted that use of the Cdc13-Cdc2 fusion does not change the order of nuclear activation and cytoplasmic activation. The phosphoproteomics was done in a Cdc13-Cdc2 fusion background and is consistent with the sensor data done in wild type cells, suggesting cyclins other than Cdc13 do not significantly contribute to the order of nuclear activation and cytoplasmic activation.

We do not think this affects the general conclusions/interpretations of the manuscript, as the intrinsic sensitivity to net CDK activity (dictated by the amino acid sequence) of the sensor does not change over the cell cycle. For example, CytCDK has a reduced sensitivity to net CDK activity prior to mitosis but will also have reduced sensitivity in mitosis. Likewise, CytCDK V2 has increased sensitivity prior to mitosis but will also have increased sensitivity in mitosis. Therefore, we believe the conclusions made using the phase plots generated with CytCDK V2 for instance, are valid, including in the background of the Cdc13-Cdc2 fusion protein, where we can be confident that we are looking at the activity of Cdc13-Cdc2 only, when plotting against Cdc13-Cdc2 levels.

5. None of the cell cycle time courses are marked with landmark events, which would be helpful for a general reader. The text often mentions these events (SPB separation, etc.) but they are not indicated.

We have now included the landmark events, SPB separation and nuclear division, in the cell cycle time traces in Figure 3, with orange and yellow circles, respectively and highlighted in the figure caption. This will help the general reader understand the ordering of mitotic events that we can identify using data from Figure 3: nuclear CDK activation, nuclear export, cytoplasmic CDK activation, SPB separation, and nuclear division.

Referees' comments:

Referee #1 (Remarks to the Author):

The authors have improved the paper in some small ways, but the two major limitations of the paper remain and have not been mitigated. The first is the issue of how much of a time lag there is between changes in Cdk1 activity and changes in the probes' translocation, and whether the time lags for the two probes are the same or different. The authors point out that the time lag could be as little as a few seconds, and that is true. Or it could be 7 min, as Spencer et al. found for their translocation probe in mammalian cells (Spencer et al. Cell 2013), or.... You can't tell without making the measurement. Measuring a translocation reporter's response to an inhibitor is a basic essential control experiment, both for assessing the specificity of the reporter (see, for example, Regot et al. Cell 2014 and Schwarz et al. Mol Cell 2018) and, in the present case, for determining the dynamics of the probe's response. In the rebuttal the authors mention that the CellAsics plates they used are not compatible with use of the Nmpp1 inhibitor, but surely there are other ways to carry out the experiment. For example, Hauf's lab used lectin-coated Ibidi dishes to immobilize *S. pombe* for fluorescence microscopy (Kamenz and Hauf Curr Biol 2014). The authors also point out that they do not have experience with other small molecule Cdk inhibitors, but why not try them? And finally, the authors point out that the re-examined proteomics data are consistent with late activation of cytoplasmic Cdk1. This is true, but it is the translocation reporter data that most directly makes the point that it is the timing of nuclear vs. cytoplasmic Cdk1 activation rather than differences in the concentrations of nuclear vs. cytoplasmic

Cdk1-Cdc13 and/or the properties of the nuclear vs. cytoplasmic substrates that makes the nuclear substrate phosphorylations occur earlier than the cytoplasmic substrate phosphorylations. The authors need to do the control experiments; without the controls, the translocation reporter experiments are not sufficiently definitive.

In our original manuscript we did not fully appreciate the importance of knowing the extent of possible nuclear/cytoplasmic translocation delays after phosphorylation for our CDK sensors. The Reviewer was quite correct to raise this issue because it is important for the conclusions we came to concerning the timings of mitotic CDK activation in the nucleus and cytoplasm. We have now carried out several experiments addressing this issue raised by the Reviewer concerning the timing of mitotic CDK activation. We thank the Reviewer for insisting that this issue is important for our work.

To address translocation delays for NucCDK and CytCDK, we set up experiments using soybean lectin-coated Mattec dishes, carrying out washes (3x) on dishes on the microscope. The shortest time interval for timelapse-acquisition we could achieve was 3 minutes – this was the fastest we could perform 3 washes carefully, and reset the microscope focus before the next timepoint. We blocked cells in G2 using 1 μ M 1-Nmpp1, released them into mitosis after washing out 1-NmPP1, and then acutely inhibited CDK activity by addition of 10 μ M 1-Nmpp1 during the rise of CDK activity in mitosis. This block and release from G2 protocol was used because CytCDK does not show readout of activity until mitotic onset. Shown below is the data obtained along with representative traces for a Cdc13-Cdc2as fusion protein strain with both NucCDK and CytCDK, to visualize activity in the nucleus and cytoplasm simultaneously.

This data is now presented in Figure 2e and Extended Data Figure 2 d, e, and described in lines 126-141.

The *change* in the rate of the sensor readout was determined, to identify when CDK activity transitions from the low activity state to the high activity state, which we

describe in the text as “mitotic CDK activation”. NucCDK has a switch-like rise immediately after the G2 release during the rise in CDK activity, and there is a delay of around 6 minutes before CytCDK begins to rise, characterized in the DMSO control, similar to the 5-10 minute delay we had reported. After adding 10uM 1-Nmpp1, NucCDK begins to drop immediately, consistent with a short time-lag of translocation after changes in CDK activity, and there is a delay of around 3 minutes between when the NucCDK readout begins to drop and when the CytCDK begins to drop. However, this delay is after CDK *inhibition* and possibly may not reflect the delay from CDK phosphorylation of the sensor, to sensor translocation. Therefore, we repeated the experiments but instead of adding 10uM 1-Nmpp1, we added 200nM 1-Nmpp1 to inhibit CDK activity but still keep cells in the mitotic state, and subsequently washed and released them after 4 timepoints. The data shown below of two representative traces indicates there is no detectable delay between the rise of NucCDK and CytCDK after release from a 200nM arrest, suggesting that there is no significant delay in the translocation of the sensors after CDK phosphorylation. Therefore, this supports the conclusion that there is a sequential ordering between nuclear activation and cytoplasmic activation at mitotic onset with a time delay of 5-10 minutes. The 3-minute delay after inhibition is also unlikely to significantly influence our results since the time interval of data acquisition was 5 minutes.

This data is now included in Figure 2f and described in lines 133-136.

We performed additional experiments to strengthen the argument that there is a sequential ordering in the rise of nuclear and cytoplasmic CDK activity at mitotic onset. We assumed that a critical CDK activity threshold needs to be reached in the nucleus to begin Cdc13 export creating a time-lag, similar to what has been reported in human cells where mitotic events are triggered at CDK activity thresholds (Gavet, O., and Pines. J, Dev.Cell (2010)). With this assumption, a more gradual rise in CDK activity in the nucleus would result in a longer time being required to reach this critical CDK threshold in the nucleus and therefore would lead to an extended time between the increase in NucCDK readout and the subsequent increase in CytCDK readout. We tested this using the Cdc13-Cdc2AFas fusion protein, where the CDKY15 feedback loops are abolished in order to make the rise in CDK activity more gradual.

The increase in NucCDK readout activity was observed to be more gradual, consistent with a slower rise in activity in the nucleus, and we also observed that there was a longer delay between the rise in NucCDK and CytCDK (characterized in the DMSO control), resulting in an average delay of 13 minutes. This was around double the time delay measured in the Cdc13-Cdc2as background. When we inhibited CDK activity, the decay half-lives and time-lags were nearly identical to the previous estimates. This

strengthens support for the view that there is a sequential ordering in the rise of nuclear CDK activity which occurs before the rise in cytoplasmic CDK activity at mitotic onset, leading to differences in the timing of CDK activation. This result supports the conclusion that mitotic CDK activation occurs first in the nucleus and then in the cytoplasm. These additional experiments are now described in lines 155-168 and the data is included in Figure 2i, and Extended Data Figures 2i,j.

As requested by the Reviewer we also characterized the half-life of decay for NucCDK and CytCDK activities in both the wild type and AF backgrounds, and found the half-lives in both backgrounds were similar at 2 minutes and 4 minutes (a difference of ~2 minutes), for NucCDK and CytCDK respectively. This indicates that the readouts for both sensors decay rapidly with fairly equal translocation kinetics. Phosphatase activity could also contribute to these decay rates. In an earlier paper from the lab we had previously measured the median decay half-lives of CDK phosphosites after CDK inhibition and found it to be around 2 minutes (Swaffer M et al (2016)). Therefore, we conclude that both CDK sensors have fast and fairly equal decay half-lives after acute CDK inhibition. This data is described in lines 139-141 and Extended Data Figures 2 e, j.

As a further test of the delay in cytoplasmic mitotic CDK activation compared with the nucleus we examined the timing of phosphorylation of T19, (which is responsible for the translocation of CytCDK) using our phosphoproteomics data set. CytCDK (synCut3) is based on a N-terminal fragment of the endogenous Cut3 protein and exhibits similar translocation properties as the endogenous version (Patterson et al (2021), eLife). The phosphorylation of T19 (in red) is similar to other cytoplasmic late substrates (in blue),

which we have shown in the manuscript (Fig 1) to rise after the nuclear substrates. This phosphoproteomics data indicates that the phosphorylation of T19 occurs after mitotic nuclear CDK activation.

These experiments using the different methods of single cell CDK sensors and phosphoproteomics address control issues raised by the Reviewer, and support the conclusion that there is a sequential order between CDK activation in the nucleus and in the cytoplasm, with mitotic CDK activation occurring first in the nucleus and then in the cytoplasm.

The second is the issue of whether the two translocation probes, one based on Cut3 and one based on Mcm2-3, are equally sensitive to Cdk activation. The authors point out that they have already made two different probes (CytCDK and CytCDK V2) with different sensitivities, and yet these probes do not seem to differ in terms of their rate change points. This is true, but it is also true that the properties of a kinase substrate can influence the timing of its phosphorylation, even if it doesn't seem to for these two particular substrates. The Nurse lab's own previous work (e.g. Swaffer et al. Cell 2016) supports the importance of the properties of the substrate, and extensive work in

budding yeast from Morgan, Loog, Uhlmann and other does as well. Given that the present paper's most important conclusions and most direct experimental approaches rely on the implicit assumption that the two translocation reporters respond rapidly and essentially identically to changes in the local Cdk1 activity, this assumption needs to be tested more thoroughly.

We agree with the Reviewer that the properties of a substrate are critical determinants of sensitivity to CDK activity and could impact our conclusions on the differences in Cdc13 concentration thresholds for CDK activation in the nucleus and cytoplasm from Figure 3f. As requested, we have now characterized the sensitivities of the sensors, NucCDK, and CytCDK. This was done by determining their *in vivo* dose-responses to CDK inhibition using 1-Nmpp1, similar to how we characterized the sensitivity of substrates (Swaffer et al, Cell (2016)). In this assay, a less sensitive substrate is one that begins to become phosphorylated only at lower concentrations of 1-Nmpp1, and thus requires higher levels of CDK activity for it to be phosphorylated. We used the Cdc13-Cdc2AFas mutant to eliminate potential non-linear responses in phosphorylation due to CDKY15 feedback loops. Cells were blocked in G1 with 5 μ M 1-NmPP1 for 1.5 generations, and were then washed and released into varying concentrations of 1-NmPP1. As shown below, there is a staggered pattern in increase between the two sensors, which suggests that CytCDK is less sensitive to CDK activity because it requires a lower concentration of 1-NmPP1 for phosphorylation to start increasing in a linear manner. This concentration is around 0.5 μ M for CytCDK and around 3 μ M for NucCDK.

We had also constructed a more sensitive version of CytCDK, CytCDK V2. CytCDK V2 was constructed by mutating T19 on CytCDK to S19 to make it more sensitive to CDK activity, using the principles from Loog and Uhlmann that the Reviewer cites and that we had referenced. Work from the Loog lab has reported that CDK phosphorylates serine better than threonine (Ord et al. Nat Struct Bio, 2019), and the Uhlmann lab reported that PP2A dephosphorylates threonine preferentially over serine (Godfrey et al, Mol Cell, 2017). These two factors should make CytCDK V2 more sensitive to net CDK activity (CDK activity and phosphatase activity). We tested this by carrying out the sensitivity assay described earlier, which showed that the sensitivity of the sensor increased and was similar to NucCDK (see below).

We have now included these figures in Extended Figure 2 f,g and have added lines 143-149.

The conclusions regarding the Cdc13 concentration thresholds at CDK activation were based on Cdc13-sfGFP intensity measurements in the nucleus and cytoplasm at the rate change points in sensor readout/substrate phosphorylation, representing mitotic CDK activation, of NucCDK and CytCDK, respectively. The change in rate of the phosphorylation/sensor readout represents a transition point in CDK activity from a low state to a high state at mitotic onset. While substrates may differ in their sensitivity to CDK activity this has less impact upon their ability to detect rate changes in mitotic CDK activation. As shown below the observed delay in CDK activation in the cytoplasm compared to the nucleus is the same for CytCDK and CytCDK V2. Therefore, CytCDK has enough sensitivity to accurately detect local mitotic CDK activation in the cytoplasm. This supports the conclusions regarding the differences in Cdc13 concentration thresholds in the nucleus and cytoplasm for mitotic CDK activation.

The Reviewer also rightly pointed out that these are just a couple of substrates, and therefore we reanalysed our phosphoproteomics data to investigate nuclear substrates

of different classes of substrate sensitivity during the cell cycle: early (high), mid (intermediate), and late (low).

Early, mid, and late phosphosites in the nucleus begin to rise at different points in the cell cycle, presumably reflecting their differences in CDK sensitivities, but they all have their 'mitotic' rate change point in phosphorylation at around 60 minutes. The rate change points of phosphosites located in the nucleus across a wide range of sensitivities to CDK activity, occur prior to the rate changes in cytoplasmic late phosphosites.

We also used the phosphoproteomics data to test more generally if there was a correlation between the sensitivity of the substrate (IC50 value) and the timing of its rate change in phosphorylation reflecting local mitotic CDK activation. This was compared with the nuclear and cytoplasmic locations of the substrates. We plotted the *in vivo* IC50 values of the data presented in Figure 1a and Extended Data Figure 1a, versus the timing of the rate change points calculated in Figure 1a.

The plot indicates that although the phosphosites have variable IC50 values, the timing of their rate change points in phosphorylation are more similar for phosphosites located together within either the nuclear or cytoplasmic spatial compartments. We conclude that the location in the spatial compartments of the nucleus and the cytoplasm is more closely associated with the timing of the rate change point rather than IC50.

We have included this data in Extended Figure 1b and described it in lines 90-93.

While most of the conclusions from the paper were based around the rate changes detected by the sensors, Figure 4 contained phase plots, so we also tested if sensitivity of the sensor changes the phase plots shown in Figure 4. We performed experiments in

the Cdc13-Cdc2-mNG background with either CytCDK and CytCDK V2 sensors and plotted the cytoplasmic phase plots (see below).

We found there was no significant difference in the phase plots using CytCDK or using the more sensitive CytCDK V2, indicating that the sensitivity of the sensor to CDK activity does not appear to affect the conclusions made in the manuscript concerning both Cdc13 concentration thresholds at CDK activation and the shapes of the phase plots.

I still find the data interesting and the conclusions of the paper plausible, but without what I consider to be essential control experiments I do not support publication.

We thank the Reviewer for their comments and advice, and hope that our new control experiments and analyses have satisfied their concerns.

Referee #2 (Remarks to the Author):

I thank the authors for providing a thorough and compelling response to my comments.

I have no further concerns.

We are happy to hear that Reviewer has no further concerns.